# Adversarial Unlearning of Backdoors via Implicit Hypergradient

Yi Zeng *[1], Si Chen[1], Won Park[2], Z. Morley Mao[2], Ming Jin[1] and Ruoxi Jia[1]

[1]Virginia Tech, Blacksburg, VA 24061, USA
[2]University of Michigan, Ann Arbor, MI 48109, USA

## Abstract

We propose a minimax formulation for removing backdoors from a given poisoned model based on a small set of clean data. This formulation encompasses much of prior work on backdoor removal. We propose the Implicit Backdoor Adversarial Unlearning (I-BAU) algorithm to solve the minimax. Unlike previous work, which breaks down the minimax into separate inner and outer problems, our algorithm utilizes the implicit hypergradient to account for the interdependence between inner and outer optimization. We theoretically analyze its convergence and the generalizability of the robustness gained by solving minimax on clean data to unseen test data. In our evaluation, we compare I-BAU with six state-of-art backdoor defenses on eleven backdoor attacks over two datasets and various attack settings, including the common setting where the attacker targets one class as well as important but underexplored settings where multiple classes are targeted. I-BAU's performance is comparable to and most often significantly better than the best baseline. Particularly, its performance is more robust to the variation on triggers, attack settings, poison ratio, and clean data size. Moreover, I-BAU requires less computation to take effect; particularly, it is more than $13\times$ faster than the most efficient baseline in the single-target attack setting. Furthermore, it can remain effective in the extreme case where the defender can only access 100 clean samples—a setting where all the baselines fail to produce acceptable results.

## 1 Introduction

In backdoor attacks, adversaries aim to embed predefined triggers into a model during training time such that a test example, when patched with the trigger, is misclassified into a target class. For instance, it has been shown that one could use a sticker as the trigger to mislead a road sign classifier to identify STOP signs to speed limit signs (Gu et al., 2017). Such attacks pose a great challenge to deploy machine learning in mission-critical applications (Li et al., 2020c; 2021a).

Various approaches have been proposed to remove the effect of backdoor attacks from a poisoned model. One popular class of approaches (Wang et al., 2019; Chen et al., 2019; Guo et al., 2019) is to first synthesize the trigger patterns from the model and then unlearn the triggers. However, these approaches presume that backdoor triggers only target a small portion of classes, thereby becoming ineffective when many classes are targeted by the attacker. Moreover, they suffer from high computational costs, as they either require synthesizing the trigger for each class independently or training additional models to synthesize the triggers all at once. Another line of works does not rely on trigger synthesis; instead, it directly mitigates the triggers' effects via, for example, fine-tuning and pruning the model parameters (Liu et al., 2018a) and preprocessing the model input (Qiu et al., 2021). While these approaches are more efficient than the trigger-synthesis-based approaches, they cannot maintain a good balance between robustness and model accuracy due to the unawareness of potential triggers. Recent work (Li et al., 2020b) proposes to leverage a teacher model to guide fine-tuning. However, our empirical studies find that the effectiveness of this approach is particularly sensitive to the underlying attack and data augmentation techniques utilized to enrich the fine-tuning dataset. Overall, it remains a challenge to design a defense that achieves a good balance between model accuracy, robustness, and computational efficiency.

To address the challenge, we propose a minimax formulation to remove backdoor triggers from a poisoned model. The formulation encompasses prior works on trigger synthesis-based defense, which solves the inner and outer problems independently. Moreover, the formulation does not make

---

*Correspondence to yizeng@vt.edu. Codes of implementations is opensourced on Github: I-BAU.

any assumption about the backdoor trigger beyond a bounded norm constraint, thus remaining effective across various attack settings. To solve the minimax, we propose an Implicit Bacdoor Adversarial Unlearning (I-BAU) algorithm based on implicit hypergradients. Unlike previous work, which breaks down the minimax into separate inner and outer optimization problems, our algorithm derives the implicit hypergradient to account for the interdependence between inner and outer optimization and uses it to update the poisoned model. We theoretically analyze the convergence of the proposed algorithm. Moreover, we investigate the generalization error of the minimax formulation, i.e., to what extent the robustness acquired by solving the minimax generalizes to unseen data in the test time. We present the generalization bounds for both linear models and neural networks. We conduct a thorough empirical evaluation of I-BAU by comparing it with six state-of-art backdoor defenses on seven backdoor attacks over two datasets and various attack settings, including the common setting where the attacker targets one class and an important but underexplored setting where multiple classes are targeted. I-BAU's performance is comparable to and most often significantly better than the best baseline. Particularly, its performance is more robust to the variation on triggers, attack settings, poison ratio, and clean data size. Moreover, I-BAU requires less computation to take effect; particularly, it is more than $13\times$ faster than the most efficient baseline in the single-target attack setting. It can still remain effective in the extreme case where the defender can only access 100 clean samples—a setting where all the baselines fail to produce acceptable results.

## 2 RELATED WORK

**Backdoor Attacks.** Backdoor attacks evolve through three stages. **1)** *Visble triggers* using unrelated patterns (Gu et al., 2017; Chen et al., 2017), or optimized triggers (Liu et al., 2018b; Bagdasaryan & Shmatikov, 2021; Zhao et al., 2020; Garg et al., 2020) for higher attack efficacy. While this line's attack methods achieve high attack success rates and clean accuracy, the triggers are visible and thus easily detected by human eyes. **2)** *Visually invisible triggers* via solving bilevil optimizations regarding the $l_p$ norm (Li et al., 2020a), or adopting nature optical effects (Liu et al., 2020; Nguyen & Tran, 2021), and *Clean label attacks* via feature embeeding (Turner et al., 2019; Saha et al., 2020) for better stealthiness. For most work from this line, the triggers or the poisons can evade the detection from manual efforts. **3)** *Invisible triggers considering other latent spaces*, e.g., the frequency domian (Zeng et al., 2021; Hammoud & Ghanem, 2021) or auto-encoder feature embedding (Li et al., 2021b), which improved the stealthiness by breaking the fundamental assumptions of a list of defenses. This paper incorporates successful attacks from these major groups and we show our method provides an effective, and generalizable defense.

**Backdoor Defenses.** Backdoor defenses can be divided into four categories: **1)** *Poison detection* via outlier detection regarding functionalities or artifacts (Gao et al., 2019; Chen et al., 2018; Tran et al., 2018; Koh & Liang, 2017; Chou et al., 2020; Zeng et al., 2021), which rely on the modeling of clean samples' distribution. **2)** *Poisoned model identification* identifies if a given model is backdoored or not (Xu et al., 2019; Wang et al., 2020). **3)** *Robust training* via differential privacy (Du et al., 2019; Weber et al., 2020) or ensembled/decoupled pipeline (Levine & Feizi, 2020; Jia et al., 2020; 2021; Huang et al., 2022). This line of work tries to achieve general robustness to withstand outlier's impact, but may suffer from low clean accuracy. **4)** *Backdoor removal* via trigger synthesising (Wang et al., 2019; Chen et al., 2019; Guo et al., 2019), or preprocessing & finetuning (Li et al., 2020b; Borgnia et al., 2020; Qiu et al., 2021). This line of work serves as the fundamental solution given a poisoned model, but there is still no satisfying solution to attaining robust results across different datasets and triggers. In this work, we propose a minimax formulation and a solver for backdoor removal. We attempt to include all cutting-edge methods under this category for comparison.

## 3 PROBLEM FORMULATION

**Attack model.** We assume that an adversary carries out a backdoor attack against a clean training set generated from the distribution $\mathcal{D}$. The adversary has pre-determined triggers and target classes. The adversarial goal is to poison the training dataset such that, given a clean test sample $x$, adding a backdoor pattern $\delta$ to $x$ (i.e., $x + \delta$) will alter the trained classifier output to be a target class $\tilde{y}$. The norm of $\delta$ represents the adversary's manipulative power. We assume that $||\delta|| \leq C_\delta$. In general, the attack can add $r$ backdoored examples into the training set to induce the association between the trigger $\delta$ and the target class $\tilde{y}$. The ratio of the total backdoored examples over the size of the training set is defined as the *poison ratio*. For better stealthiness, attacks often aim not to affect the prediction of a test example when the trigger is absent. Note that in our attack model, we explicitly acknowledge the possibility that multiple trigger patterns and multiple target classes are present. We will refer to the model trained on the poisoned training set as a *poisoned model*.

**Defense goal.** We consdier that the defender is given a poisoned classifier $f_{\theta_{\text{poi}}}$ and an extra set of clean data $D = \{(x_i, y_i)\}_{i=1}^{n}$ from $\mathcal{D}$. With $f_{\theta_{\text{poi}}}$ and $D$, the defender aims to build a model $f_{\theta^*}$ immunne to backdoors, i.e., $f_{\theta^*}(x + \delta) = f_{\theta^*}(x)$. Note that the size of available clean examples is assumed to be much smaller than the size necessary to retrain a high-accuracy model from scratch.

**Minimax formulation for defense.** To achieve the defense goal, we want the resulting classifier to maintain the correct label even if the attacker patches the backdoor trigger to a given input. This intuition naturally leads to the following minimax optimization formulation of backdoor removal:

$$\theta^* = \arg\min_{\theta} \max_{\|\delta\| \leq C_\delta} H(\delta, \theta) := \frac{1}{n} \sum_{i=1}^{n} L(f_\theta(x_i + \delta), y_i), \tag{1}$$

where $L$ is the loss function. Note that this formulation looks similar to adversarial training for evasion attacks (a.k.a. adversarial examples) (Madry et al., 2017). The main distinction is that in evasion attacks, the adversarial perturbation is specific to an individual data point, while in backdoor attacks, the same perturbation (i.e., the backdoor trigger) is expected to cause misclassifications for *any* examples upon being patched. Hence, in the formulation for backdoor attacks, the same trigger, $\delta$, is applied to every clean sample $x_i$, whereas, in the minimax formulation for evasion attacks, the perturbation is optimized independently for each sample (see Eq. (2.1), Madry et al. (2017)).

The minimax formulation for backdoor removal has many advantages. *1)*, it gives us a unifying perspective that encompasses much prior work on backdoor removal. The formulation comprises an inner maximization problem and an outer minimization problem. Both of these problems have a natural interpretation in the security context. The inner maximization problem aims to find a trigger that causes a high loss for predicting the correct label. This aligns with the problem of a backdoor attack that misleads the model to predict a predefined incorrect target label. In our formulation, we choose to maximize the prediction loss for the correct label instead of using the exact backdoor attack objective of minimizing the prediction loss associated with the target label. We make this design choice because, in reality, the defender does not know the target labels selected by the attacker. The outer minimization problem is to find model parameters so that the "adversarial loss" given by the inner attack problem is minimized. Existing backdoor removal approaches based on trigger synthesis and unlearning the synthesized trigger (Wang et al., 2019; Chen et al., 2019; Guo et al., 2019) can be considered as a special instance of this minimax formulation, where they neglect the interdependence between the inner and outer optimization and solve them separately. *2)*, the formulation does not make any assumption on the backdoor trigger beyond the bounded norm constraint. By contrast, much of the existing work makes additional assumptions about backdoor triggers in order to be effective. For instance, synthesis-based approaches often assume that the classes coinciding with the target classes of the triggers account for only a tiny portion of the total, making those approaches ineffective in countering multiple target cases. *3)*, the formulation provides a quantitative measure of backdoor robustness. In particular, when the parameters $\theta$ yield a (nearly) vanishing loss, the corresponding model is perfectly robust to attacks specified by our attack model on $D$. We will discuss the generalization properties of this formulation in Section 5. Using the results developed there, we can further reason about the robustness on the data distribution $\mathcal{D}$.

## 4 ALGORITHM

**A natural (but problematic) algorithm design.** Given the empirical success of adversarial training for defending against evasion attacks (Li et al., 2022), one may wonder whether we can solve the minimax for the backdoor attack via adversarial training. Adversarial training keeps alternating between two steps until the loss converges: (1) solving the inner maximization for a fixed outer minimization variable; and (2) solving the outer minimization by taking a gradient of the loss evaluated at the maximum. Practically, adversarial training proceeds by first generating adversarial perturbations and then updating the model using the gradient calculated from the perturbed data. For backdoor attacks, the adversarial perturbation needs to fool the model *universally* on all inputs. Hence, a natural algorithm to solve the backdoor minimax problem is to tune the model with universal adversarial perturbations. Universal adversarial perturbations have been studied in the past, and there exists an off-the-shelf technique to create such perturbations (Moosavi-Dezfooli et al., 2017). However, we found that this simple algorithm is highly unstable. Figure 1 illustrates the change of the model accuracy and robustness (measured in terms of the attack success rate (ASR)) for 20 runs as adversarial training proceeds. It can be seen that the ASR varies significantly across different runs. Within a single run, while the ASR decays as a whole, the decrement is mostly erratic and slow.

There are two issues with naive adversarial training with universal perturbations. First, the performance of the existing universal perturbation algorithm is unstable. At its core, it generates adversarial perturbations for each example and adds the perturbations together. The addition may not always lead to a perturbation that can fool all examples; instead, the addition operation may cancel the effect of individual perturbations. More importantly, the decoupling between solving the inner maximization and the outer minimization in adversarial training is often justified by Danskin's Theorem (Danskin, 2012), which states that the gradient of the inner function involving the maximization term is simply given by the gradient of the function evaluated at this maximum. In other words, letting $\delta^* = \arg\max_{||\delta|| \leq C_\delta} H(\delta, \theta)$, we have $\nabla_\theta \max_{||\delta|| \leq C_\delta} H(\delta, \theta) = \nabla_\theta H(\delta^*, \theta)$. However, in practice, due to stochastic gradient descent, it is impossible to solve the inner maximization optimally. Worse yet, the underperformance of the existing universal perturbation algorithm makes it even harder to approach the maximum. Moreover, Danskin's Theorem only holds for convex loss functions. Due to the violation of maximum condition and convexity, it is unsuitable for utilizing Danskin's Theorem, and hence, adversarial training is not justified.

---

**Algorithm 1:** Implicit Backdoor Adversarial Unlearning (I-BAU)

**Input:** $\theta_1$ (poisoned model); $\quad$ $D$ (clean set accessible for backdoor unlearning);
**Output:** $\theta_K$ (sanitized model)
**Parameters:** $C_\delta$ ($\ell_2$ norm bound); $\quad$ $K, T$ (outer and inner iteration numbers); $\quad$ $\alpha, \beta > 0$ (step sizes)

1 **for** *each iteration* $i \in (1, K-1)$ **do**
2 $\quad$ $\delta_i \leftarrow \mathbf{0}^{1 \times d}$;
3 $\quad$ **for** $t \in (1, T)$ **do**
4 $\quad\quad$ Update $\delta_i^{t+1} = \delta_i^t + \alpha \nabla_1 H(\delta_i^t(\theta_i), \theta_i)$;
5 $\quad\quad$ $\delta_i = \delta_i \times \min\left(1, \frac{C_\delta}{\|\delta_i\|_2}\right)$;
6 $\quad$ Compute $\nabla \delta_i^\top$ (Hessian products) via an iterative solver with reverse mode differentiation;
7 $\quad$ $\nabla \tilde{\psi}(\theta_i) = \nabla_2 H(\delta_i, \theta_i) + \nabla \delta_i^\top \nabla_1 H(\delta_i, \theta_i)$;
8 $\quad$ Update $\theta_{i+1} = \theta_i - \beta \nabla \tilde{\psi}(\theta_i)$;
9 **return** $\theta_K$

---

**Proposed algorithm.** We propose an algorithm to solve the minimax problem in (1) that does not require the loss function to be convex and is more robust to the approximation error caused by not being able to solve the inner maximization problem to global or even local optimality. Let $\delta(\theta)$ be an arbitrary suboptimal solution to $\max_{||\delta|| \leq C_\delta} H(\delta, \theta)$. Let $\psi(\theta) \coloneqq H(\delta(\theta), \theta)$ be the objective function evaluated at the suboptimal point; if $\delta(\theta)$ is a stationary point, we make an distinction by defining the corresponding function as $\psi^*(\theta)$. Denote $\nabla_1 H(\delta, \theta)$ and $\nabla_2 H(\delta, \theta)$ as the partial derivatives with respect to the first and second variable, respectively, and $\nabla_1^2 H(\delta, \theta)$ and $\nabla_{1,2}^2 H(\delta, \theta)$ as the second-order derivatives of $H$ with respect to the first variable and the mixed variables, respectively. The gradient of $\psi$ with respect to $\theta$ is given by

$$\underbrace{\nabla \psi(\theta)}_{\text{hypergrad. of } \theta} = \underbrace{\nabla_2 H(\delta(\theta), \theta)}_{\text{direct grad. of } \theta} + \underbrace{\overbrace{(\nabla \delta(\theta))^\top}^{\text{response Jacobian}} \overbrace{\nabla_1 H(\delta(\theta), \theta)}^{\text{direct grad. of } \delta}}_{\text{indirect grad. of } \theta}. \tag{2}$$

The direct gradients are easy to compute. For instance, suppose $H$ is the loss of a neural network, then $\nabla_1 H(\delta(\theta), \theta)$ and $\nabla_2 H(\delta(\theta), \theta)$ are the gradients with respect to the network input and the model parameters, respectively. However, the response Jacobian is intractable to obtain because we must compute the change rate of the suboptimal solution to the inner maximization problem with respect to $\theta$. When $\delta(\theta)$ satsifies the first-order stationarity condition, i.e., $\nabla_1 H(\delta(\theta), \theta) = 0$, and assuming that $\nabla_1^2 H(\delta(\theta), \theta)$ is invertible, the response Jacobian is given by

$$\nabla \delta(\theta) = -\left(\nabla_1^2 H(\delta(\theta), \theta)\right)^{-1} \nabla_{1,2}^2 H(\delta(\theta), \theta), \tag{3}$$

which follows from the implicit function theorem. Note that $\nabla \delta(\theta)$ plays a role in the Hessian in (2), in the sense that it adjusts each dimension of the gradient $\nabla_1 H(\delta(\theta), \theta)$ to the importance of that dimension. Hessian expresses the importance via curvature, whereas $\nabla \delta(\theta)$ measures the importance based on the sensitivity to the change of $\theta$. A widely known fact from the optimization literature is that second-order optimization algorithms are tolerant to the inaccuracy of Hessian (Byrd et al., 2011). Based on this intuition, we propose to approximate $\nabla \delta(\theta)$ with a suboptimal solution by (3). In practice, the approximation is addressed with an iterative solver of limited rounds (e.g., conjugated gradient algorithm (Rajeswaran et al., 2019) or fixed-point algorithm (Grazzi et al., 2020)) along with the reverse mode of automatic differentiation (Griewank & Walther, 2008).

Finally, to solve (1), we plug in the approximate response Jacobian into (2). Then, we use $\nabla\tilde{\psi}(\theta)$ (referred to as *implicit hypergradient* hereinafter) to perform gradient descent and update the poisoned model. Note that the implicit hypergradient only depends on the value of $\delta(\theta)$ instead of its optimization path; hence, it can be implemented in a memory-efficient manner compared to the explicit way of solving bi-level optimizations (Grazzi et al., 2020). The overall algorithm, dubbed *Implicit Backdoor Adversarial Unlearning*, for $\ell_2$-norm bounded attacks is presented in Algorithm 1. Generalizing the algorithm to other norms is straightforward. We found that imposing a large norm bound in I-BAU has no discernible effect on clean accuracy (Appendix A.3.1). This empirical observation enables the proposed I-BAU to be flexible enough to deal with different scales of backdoor triggers in practice. Further details on the the iterative solver with the suboptimum solution and memory complexity analysis are provided in Appendix A.3.

## 5 THEORETICAL ANALYSIS

In this section, we analyze the convergence of the I-BAU. We also study the generalization properties of the minimax backdoor removal formulation. Assuming that by solving the minimax, we obtain a model such that all points in the clean set $D$ are robust to triggers of a specific norm, we would like to reason about to what extent an unseen point from the underlying data distribution $\mathcal{D}$ is trigger-robust.

**Convergence Bound:** Suppose that $H(\cdot,\theta)$ is $\mu_H(\theta)$-strongly convex and $L_H(\theta)$-Lipschitz smooth, where $\mu_H(\cdot)$ and $L_H(\cdot)$ are continuously differentiable. Define $\alpha(\delta) = \frac{2}{L_H(\theta)+\mu_H(\theta)}$, and let $\delta(\theta)$ be the unique fixed point of $\delta + \alpha(\theta)\nabla_1 H(\delta,\theta)$. Then, there exists $L_{H,\theta}$ s.t. $\sup_{\|\delta\|\le 2C_\delta}\|\nabla_1 H(\delta,\theta)\| \le L_{H,\theta}$ (Grazzi et al., 2020). We make the following assumptions:

- *Lipschitz continuity of direct gradients*: $\nabla_1 H(\cdot,\theta)$ and $\nabla_2 H(\cdot,\theta)$ are Lipschitz continuous functions of $\theta$ (with Lipschitz constants $\eta_{1,\theta}$ and $\eta_{2,\theta}$, repectively).
- *Lipschitz continuity and norm boundedness of second-order terms*: $\nabla_1^2 H(\cdot,\theta)$ is $\hat{\rho}_{1,\theta-}$ Lipschitz continuous, and $\nabla_{2,1}^2 H(\cdot,\theta)$ is $\hat{\rho}_{2,\theta}$-Lipschitz continuous. Also, the cross derivative term is norm bounded by $L_{h,\theta} \ge \left\|\nabla_{21}^2 H(\delta(\theta),\theta)\right\|$.
- *Asymptotic convergence of $\delta^t(\theta)$ to $\delta(\theta)$*: $\|\delta^t(\theta) - \delta(\theta)\| \le \rho_\theta(t)\|\delta(\theta)\|$, where $\rho_\theta(t)$ is such that $\rho_\theta(t) \le 1$, and $\rho_\theta(t) \to 0$ as $t \to +\infty$.

**Theorem 1** *Consider the approximate hypergradient $\nabla\tilde{\psi}(\theta_i)$ computed by Algorithm 1 (line 7). For every inner and outer iterate $t, i \in \mathbb{N}$, we have:*

$$\left\|\nabla\tilde{\psi}(\theta_i) - \nabla\psi(\theta_i)\right\| \le \left(c_1(\theta_i) + c_2(\theta_i)\frac{1-q_{\theta_i}^t}{1-q_{\theta_i}}\right)\rho_{\theta_i}(t) + c_3(\theta_i)q_{\theta_i}^t, \qquad (4)$$

*where $q_{\theta_i} := \max\left\{1 - \alpha(\theta_i)\mu_H(\theta_i), \alpha(\theta_i)L_H(\theta_i) - 1\right\}$, $c_1(\theta_i) := \left(\eta_{2,\theta_i} + \frac{\eta_{1,\theta_i}L_{h,\theta_i}\alpha(\theta_i)}{1-q_{\theta_i}}\right)C_\delta$, $c_2(\theta_i) := L_{H,\theta_i}C_\delta\left(L_H(\theta_i)\|\nabla\alpha(\theta_i)\| + \hat{\rho}_{2,\theta_i}\alpha(\theta_i)\right) + \frac{\hat{\rho}_{1,\theta_i}L_{H,\theta_i}L_{h,\theta_i}C_\delta\alpha(\theta_i)^2}{1-q_{\theta_i}}$, and finally $c_3(\theta_i) := \frac{L_{H,\theta_i}L_{h,\theta_i}\alpha(\theta_i)}{1-q_{\theta_i}}$.*

As the inner epoch number $t$ increases, the estimated gradient becomes more accurate. Under the assumption that $H(\cdot,\theta)$ is convex, the solution converges. We show the full proof in Appendix A.1.

**Generalization Bound for Linear Models:** Consider a class of linear classifiers $\theta \in \Theta$ where the weights are norm-bounded by $C_\theta$. Define $\chi$ to be the upper bound of any input sample $x \in D$, i.e., $\|x\|_2 \le \chi$. Let the empirical risk defined as $\hat{R}_\gamma(\theta) \le n^{-1}\sum_j \mathbb{1}\left[\theta(x_j + \delta)_{y_j} \le \gamma + \max_{j\ne y_j}\theta(x_j + \delta)_j\right]$ with a margin $\gamma > 0$.

**Theorem 2** *For any linear model $\theta$ and $\xi \in (0,1)$, the following holds with a probability of at least $1 - \xi$ over $((x_i, y_i))_{i=1}^n$:*

$$\Pr\left[\arg\max_j [\theta(x+\delta)_j] \ne y\right] \le \hat{R}_\gamma(\theta) + 2C_\theta\frac{(\chi + C_\delta)}{\sqrt{n}} + 3\sqrt{\frac{\ln(1/\xi)}{2n}}, \qquad (5)$$

When the number of clean samples, $n$, gets larger, the generalizability of the adversarial unlearning on linear models becomes better. It is also interesting to compare the adversarial generalization bound for backdoor attacks with the bound for evasion attacks in (Yin et al., 2019), which has an additional $\sqrt{d}$ factor in the second term of the bound. Hence, for inputs of large dimensions, the adversarial training for backdoor attacks is expected to be more generalizable than that for evasion attacks. The detailed proof is shown in the Appendix A.2.1.

**Generalization Bound for Neural Networks:** Consider a neural network with $L$ fixed nonlinearities $(\sigma_1, \ldots, \sigma_L)$, where $\sigma_i$ is $\varrho_i$-Lipschitz (e.g., coordinate-wise ReLU, max-pooling as discussed in (Bartlett et al., 2017)) and $\sigma_i(0) = 0$. Let $L$ weight matrices be denoted by $(A_1, \ldots, A_L)$ with dimensions $(d_0, \ldots, d_L)$, respectively, where $d_0 = d$ is the number of input dimensions, $d_L = C$ is the number of classes, and $W = \max(d_0, \ldots, d_L)$ is the highest dimension layer. Assume that the weight matrices satisfy $\|A_i\|_{2,1} \leq a_i$, $\|A_1\|_{2,1} \leq a_0$, and $A_i \in \mathbb{R}^{d_i \times m_i}$. Let $a_1 = a_0 \left(1 + m_1 n^{1/2} C_\delta\right)$. The whole network's output regarding a backdoored sample is denoted as: $\sigma_L \left(A_L \sigma_{L-1} \left(A_{L-1} \ldots \sigma_1 \left(A_1(x + \delta)\right) \ldots\right)\right)$. Let $\|A_i\|_\sigma$ be bounded by the spectral norms $s_i$.

**Theorem 3** *For any neural network model $\theta$ and $\xi \in (0, 1)$, the following holds with a probability of at least $1 - \xi$ over $((x_i, y_i))_{i=1}^n$:*

$$
\Pr\left[\arg\max_j \left[\theta(x + \delta)_j\right] \neq y\right] \leq
$$
$$
\hat{R}_\gamma(\theta) + \frac{8}{n} + \frac{48\ln(n)(\|X\|_p + \sqrt{n})\sqrt{\ln(2W(W+n))}}{\gamma n} \left(\prod_{i=1}^L s_i \varrho_i\right) \left(\sum_{i=1}^L \frac{a_i^{2/3}}{s_i^{2/3}}\right)^{3/2} + 3\sqrt{\frac{\ln(1/\xi)}{2n}}. \tag{6}
$$

When the number of clean samples, $n$, gets larger, the generalizability of the unlearning on neural networks becomes better. Compared with the generalization bound for regular training in (Bartlett et al., 2017), our bound has an additional $n$ term in $\sqrt{\ln(2W(W+n))}$ and $(\|X\|_p + \sqrt{n})$, which indicates harder generalizability than regular learning problem. However, empirically, we find our solution, I-BAU, is still of excellent generalizability even if only 100 clean samples are accessible. Our proof is enabled by recognizing that the backdoor perturbation, $\delta$, is required to be the same across all the poison samples and therefore can be treated as additional dimensions of the model parameters. The proof implements Dudley entropy integral to bound the Rademacher complexity. The full details are deferred to Appendix A.2.2. We also empirically validate Theorem 3 against $W$ and $n$ in Appendix A.2.3.

## 6 EVALUATION

Our evaluation aims to answer the following questions: (1) Efficacy: Can I-BAU effectively remove various backdoor triggers? (2) Stability: Can I-BAU be consistently effective across different runs? (3) Sensitivity: How sensitive is I-BAU to the poison ratio and the size of the available clean set? (4) Efficiency: How efficient is I-BAU? We use a simplified VGG model (Simonyan & Zisserman, 2014) as the target model for all experiments and set $C_\delta = 10$ as the norm constraint for implementing I-BAU. The details of experimental settings and model architecture can be found in Appendix A.4.

### 6.1 EFFICACY

We evaluate I-BAU's efficacy against three attack settings: 1) One-trigger-one-target attack, in which the attacker uses only one trigger, and there is only one target label. This setting is most commonly considered in existing defense works. 2) One-trigger-all-to-all attack, in which the attacker uses only one trigger but aims to mislead predictions from $i$ to $i+1$, where $i$ is the ground truth label (Gu et al., 2017). 3) Multi-trigger-multi-target attack, where the attacker uses multiple distinct triggers, each targeting a different label. We will refer to the first two settings as "one-trigger setting" and the last setting as "multi-trigger setting." For each setting, we study seven different backdoor triggers in the main text, namely, *BadNets white square trigger* (BadNets) (Gu et al., 2017), *Hello Kitty blending trigger* (Blend) (Chen et al., 2017), $l_0$ *norm constraint invisible trigger* ($l_0$ inv) (Li et al., 2020a), $l_2$ *norm constraint invisible trigger* ($l_2$ inv) (Li et al., 2020a), *Smooth trigger (frequency invisble trigger)* (Smooth) (Zeng et al., 2021), *Trojan square* (Troj SQ) (Liu et al., 2018b), and *Trojan watermark* (Troj WM) (Liu et al., 2018b). These trigger patterns are illustrated in Figure 3 in the Appendix. Given that I-BAU's fundamental formulation takes backdoor noise as additive noise, one may be curious about how I-BAU's performance mitigates non-additive backdoor attacks. We have included case studies on using I-BAU mitigating four non-additive triggers (semantical replacement, WaNet (Nguyen & Tran, 2021), IAB attack (Nguyen & Tran, 2020)) and hidden trigger attack (Saha et al., 2020) (non-additive poisoning) to illustrate the effectiveness (Appendix A.5). For all experiments in the mian text, we adopt a large poison ratio of 20%, a severe attack case for defenders. To acquire a poisoned model, we train on the poisoned dataset for 50 epochs.

We compare I-BAU with six state-of-art defenses: *Neural Cleanse* (NC) (Wang et al., 2019), *Deepinspect* (DI) (Chen et al., 2019), TABOR (Guo et al., 2019), *Fine-pruning* (FP) (Liu et al., 2018a),

*Neural Attention Distillation* (NAD) (Li et al., 2020b), and *Differential Privacy* training (DP) (Du et al., 2019). Note that DP requires access to the poisoned data; hence, its attack model is different from the attack model of the other baselines and our method. In the following tables, we use [ ASR ] to mark the results that fail to reduce the attack success rate (ASR) below 20%; we use [ ACC ] to mark the results where the accuracy (ACC) on clean data drops by more than 10%; we use [ ACC or ASR ] to mark the best result among the six baselines; finally, we use [ ACC or ASR ] to mark the results from I-BAU that is comparable to or significantly better than the best result among the baselines. We consider I-BAU's result as comparable if **1)** the ACC gap is less than 1%, **2)** the ASR gap is less than 4%, or the ASR of I-BAU is close to the label percental (i.e., the probability of random guessing for outputting the target label, 10% for the CIFAR-10, 2.3 % for the GTSRB).

**One-trigger Setting:** Table 1 presents the defense results on the CIFAR-10 dataset. CIFAR-10 contains 60,000 samples. We use 50,000 samples as training data, among which 10,000 samples are poisoned. We used 5,000 separate samples as the clean set accessible to the defender for conducting each defense (e.g., via unlearning, finetuning, trigger synthesis) except DP. The remaining 5,000 samples are used to assess the defense result. The left column depicts the attack cases. The first seven cases each contain only one target label. The all-to-all case represents the case in which we use the BadNets trigger to carry out the one-trigger-all-to-all attack. As shown in the table, all single-target attacks are capable of achieving an ASR close to 100% with no defenses. It is empirically observed that the ASR of the all-to-all attack is upper-bounded by the ACC. Intuitively, the model needs to classify the clean samples correctly to classify the corresponding backdoored samples with triggers to the next class. Thus, we tune the model to produce an ASR that is close to the ACC.

| Attack | No Defense | | NC | | DI | | TABOR | | FP | | NAD | | DP | | I-BAU(Ours) | |
|---|---|---|---|---|---|---|---|---|---|---|---|---|---|---|---|---|
| | ACC | ASR | ACC | ASR | ACC | ASR | ACC | ASR | ACC | ASR | ACC | ASR | ACC | ASR | ACC | ASR |
| BadNets | 84.94 | 98.28 | 83.42 | 8.76 | 82.12 | 48.50 | 83.78 | 8.12 | 82.22 | 96.66 | 78.72 | 11.66 | 11.08 | 21.77 | 83.35 | 12.30 |
| Blend | 84.82 | 99.78 | 83.08 | 33.96 | 81.84 | 52.62 | 83.42 | 21.26 | 81.24 | 89.78 | 77.48 | 13.02 | 11.68 | 13.72 | 82.30 | 12.96 |
| $l_0$ inv | 85.36 | 100 | 83.02 | 8.78 | 83.40 | 29.1 | 82.27 | 8.16 | 82.18 | 100 | 65.08 | 7.22 | 12.48 | 25.22 | 84.08 | 9.54 |
| $l_2$ inv | 85.26 | 100 | 80.68 | 8.08 | 82.46 | 7.82 | 80.30 | 11.64 | 81.50 | 98.94 | 43.18 | 12.56 | 11.58 | 20.57 | 83.48 | 7.48 |
| Smooth | 85.34 | 99.24 | 83.72 | 46.88 | 83.32 | 61.82 | 84.14 | 45.94 | 82.66 | 9.44 | 77.22 | 54.38 | 10.70 | 28.14 | 83.46 | 18.30 |
| Trojan SQ | 84.76 | 99.66 | 81.30 | 8.02 | 83.14 | 6.94 | 81.38 | 7.06 | 82.34 | 99.50 | 51.86 | 7.84 | 10.70 | 18.26 | 83.18 | 9.82 |
| Trojan WM | 84.92 | 99.96 | 81.76 | 6.02 | 82.88 | 7.24 | 82.60 | 49.26 | 81.64 | 99.88 | 56.84 | 0.82 | 15.21 | 32.89 | 83.58 | 3.42 |
| All to all | 86.38 | 85.02 | 85.38 | 82.88 | 84.74 | 56.38 | × | × | 84.48 | 66.46 | 75.70 | 2.34 | 14.80 | 10.93 | 80.34 | 10.46 |

Table 1: Results on CIFAR-10, one-trigger cases. CIFAR-10's ACC is sensitive to fine-tuning and I-BAU; we compare I-BAU when it drops similar ACCs amount to the most effective method. × - no detected trigger.

Model performance on CIFAR-10 is particularly sensitive to finetuning and unlearning, which will result in a decrease in the ACC in general. Thus, for a fair comparison, we show the I-BAU results in Table 1 when the ACC falls to the same level as the **most effective** baseline. For instance, for BadNets attacks, we present the ASR result of I-BAU when the ACC of I-BAU decreases to a value similar to the ACC of TABOR, which is the best defense baseline against BadNets.

| Attack | No Defense | | TABOR | | FP | | NAD | | DP | | I-BAU(Ours) | |
|---|---|---|---|---|---|---|---|---|---|---|---|---|
| | ACC | ASR | ACC | ASR | ACC | ASR | ACC | ASR | ACC | ASR | ACC | ASR |
| BadNets | 97.69 | 99.18 | 98.99 | 4.16 | 99.34 | 60.19 | 15.19 | 9.08 | 6.46 | 100 | 99.38 | 3.32 |
| Blend | 97.44 | 99.91 | 99.09 | 33.32 | 99.52 | 69.66 | 47.71 | 18.48 | 5.70 | 100 | 98.89 | 5.01 |
| $l_0$ inv | 97.72 | 100 | 98.80 | 0.47 | 99.41 | 74.86 | 17.41 | 1.20 | 7.40 | 88.23 | 99.24 | 0.42 |
| $l_2$ inv | 97.57 | 99.91 | 98.51 | 0.41 | 99.53 | 40.46 | 15.50 | 1.16 | 5.46 | 100 | 97.75 | 0.45 |
| Smooth | 97.87 | 99.89 | 98.62 | 0.47 | 99.55 | 47.75 | 10.06 | 0.70 | 5.94 | 95.58 | 98.96 | 0.22 |
| Trojan SQ | 98.12 | 99.98 | 99.06 | 5.70 | 99.48 | 75.96 | 23.31 | 14.68 | 5.51 | 100 | 99.04 | 5.11 |
| Trojan WM | 97.84 | 100 | 98.63 | 5.40 | 99.45 | 69.82 | 11.16 | 13.62 | 5.70 | 100 | 99.44 | 2.55 |
| All to all | 97.10 | 95.42 | 98.63 | 47.07 | 99.45 | 67.34 | 25.53 | 0.42 | 5.87 | 5.70 | 99.13 | 0.04 |

Table 2: Results on GTSRB, one-trigger cases. I-BAU's results shown here were obtained after 100 rounds of I-BAU. For that, Neural Cleanse, Deppinspect, and TABOR are from the same line of work, so we here only compare the result with the most state-of-art method in this category, TABOR.

As shown in Table 1, the performance of the baselines exhibits large variance across different triggers. Specifically, each baseline underperforms in at least three attack cases (marked by ([ ASR ] or [ ACC ])) The defenses based on trigger synthesis (including NC, DI, and TABOR) failed to confront the all-to-all attack case, where all the labels were targeted. This is because this setting violates their assumption that target classes only account for the minority of training data. Particularly, TABOR fails to detect any backdoor triggers in this case). For DP, we fine-tune a noise multiplier effective to mitigate all attacks, yet also leads to bad ACC. On the other hand, I-BAU robustly mitigates all triggers without significantly affecting the ACC. Compared to the best result among the state-of-art baselines ([ ACC or ASR ]), I-BAU's is comparable to or much better ([ ACC or ASR ]) under most of the settings. The only setting where I-BAU underperforms the best baseline is Smooth triggers. FP is the only effective baseline in this setting. But interestingly, this setting is also the only setting where FP is effective; in other words, its performance is highly dependent on the underlying trigger.

| | | No Def. | NC | DI | TABOR | FP | NAD | DP | Ours* |
|---|---|---|---|---|---|---|---|---|---|
| CIFAR-10 | **ACC** | 85.96 | × | 83.74 | 84.04 | 83.42 | 77.38 | 11.18 | 77.44 |
| | **avg. ASR** | 98.37 | × | 30.37 | 40.68 | 73.18 | 10.93 | 12.83 | 12.96 |
| | Trojan WM: ⇒ 9 | 99.92 | × | 9.68 | 28.02 | 99.7 | 18.38 | 13.40 | 11.48 |
| | Trojan SQ: ⇒ 2 | 99.42 | × | 13.78 | 19.56 | 99.36 | 11.94 | 2.58 | 10.80 |
| | BadNets: ⇒ 0 | 94.5 | × | 72.6 | 9.16 | 9.56 | 13.32 | 6.94 | 18.64 |
| | Smooth: ⇒ 1 | 97.08 | × | 30.58 | 89.70 | 9.32 | 8.38 | 6.28 | 15.22 |
| | Blend: ⇒ 3 | 98.12 | × | 48.84 | 50.84 | 96.14 | 8.46 | 27.98 | 18.24 |
| | $l_0$ inv: ⇒ 4 | 100 | × | 23.78 | 80.96 | 99.88 | 9.88 | 18.02 | 9.46 |
| | $l_2$ inv: ⇒ 5 | 99.54 | × | 13.34 | 6.52 | 98.32 | 6.16 | 14.62 | 6.92 |
| GTSRB | **ACC** | 97.18 | 68.72 | 99.33 | 76.38 | 99.36 | 10.83 | 6.17 | 99.09 |
| | **avg. ASR** | 99.37 | 10.87 | 7.21 | 11.89 | 9.38 | 11.59 | 42.86 | 4.18 |
| | Trojan WM: ⇒ 9 | 99.49 | 4.00 | 3.60 | 3.86 | 1.56 | 6.61 | 100 | 1.78 |
| | Trojan SQ: ⇒ 2 | 99.59 | 2.79 | 6.26 | 2.33 | 0.12 | 4.14 | 0 | 5.68 |
| | BadNets: ⇒ 0 | 98.19 | 6.61 | 0.47 | 10.29 | 14.09 | 4.79 | 0 | 0.72 |
| | Smooth: ⇒ 1 | 99.89 | 8.56 | 6.92 | 15.28 | 10.05 | 2.83 | 100 | 5.65 |
| | Blend: ⇒ 3 | 99.15 | 46.89 | 22.66 | 45.91 | 29.69 | 41.54 | 100 | 3.76 |
| | $l_0$ inv: ⇒ 4 | 100 | 1.31 | 5.15 | 1.06 | 0.59 | 17.70 | 0 | 5.25 |
| | $l_2$ inv: ⇒ 5 | 99.33 | 5.94 | 5.43 | 4.52 | 9.60 | 3.52 | 0 | 6.46 |

Table 3: Results for 7-trigger-7-target cases. × marks no trigger was detected. *Here, ASR results on CIFAR-10 are provided when the model attained an ACC similar to that of NAD (the only effective one on CIFAR-10).

Table 2 shows the evaluation on the GTSRB dataset, which contains 39,209 training data and 12,630 test data of 43 different classes. The experimental setting is the same as the one on CIFAR, except for the data split, which is discussed in the Appendix A.4.1. We find that model performance on GTSRB is not sensitive to the tuning procedure. As 5,000 clean samples are available to the defender, the ACC of the model increases after incorporating the defense. NAD's performance highly depends on the pre-designed preprocessing procedure adopted during fine-tuning. While it produces the best defense results for some settings on CIFAR-10, it suffers from a large degradation of ACC on GTSRB. Once again, I-BAU is the only defense method that remains effective for all attack settings.

**Multi-trigger Setting:** Table 3 shows the results on CIFAR and GTSRB in a 7-trigger-7-target setting, in which each trigger targets a different label. We adopt the same poison ratio for each (20%) and then combine the seven distinct poisoned datasets to obtain the final datasets (size 350,000 for CIFAR-10, and 274,463 for the GTSRB). We show the average ASR and the specific ASR.

On CIFAR-10, since the target classes are no longer the minority (i.e., 7/10 labels are targeted), the baselines based on trigger synthesis, which make the minority assumption, are ineffective. Particularly, NC fails to detect any triggers in this setting. NAD is the only baseline able to evaluate well on CIFAR-10, and our methods achieve comparable performance. However, similar to the one-trigger setting, NAD's performance requires the customization of preprocessing to different datasets. The default preprocessing cannot maintain the same performance on GTSRB; e.g., the random flipping—a beneficial preprocessing step for CIFAR—completely alters the semantics of images in GTSRB. On the other hand, I-BAU effectively reduces ASR while maintaining the ACC for both datasets with no changes needed. Moreover, in the Appendix, we visualize the distribution of poisoned and clean examples in the feature space before and after applying I-BAU, which shows that I-BAU can place the poisoned examples back in the cluster with correct labels.

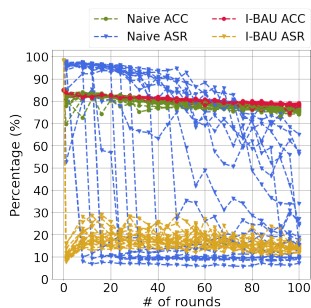

Figure 1: Comparison of Naive and I-BAU. I-BAU's performance is more stable.

## 6.2 STABILITY

In Section 4, we mentioned that the naive heuristic based on adversarial learning with existing universal perturbation suffers from erratic performance. Here, we aim to evaluate whether I-BAU can overcome this problem. We adopt the same setting for the two on the CIFAR-10 using BadNets and observe the change in ACC and ASR over different iterations for each run. As shown in Figure 1, the ASR decreases very quickly; indeed, the attack can be mitigated by just one single iteration of outer minimization. Also, within every single run, the ASR decreases more smoothly compared to the naive heuristic. Notably, I-BAU attains a slightly higher ACC than the naive heuristic.

## 6.3 SENSITIVITY

We evaluate the sensitivity of each defense to the poison ratio and the size of clean samples. The experiments were performed on Trojan WM on CIFAR-10 to exemplify the results. Table 4 shows the sensitivity to the poison ratio. Observe that as the poison ratio drops, TABOR becomes ineffective at synthesizing triggers and ultimately fails to remove the triggers; NC, however, becomes the most effective baseline. While DP's ASR is significantly worse than the effective baselines, its performance gets slightly better as the poison ratio drops. DP attempts to restrict the impact of each training data point on the learning outcome via noising the gradient. As a side effect, it hinders learning from good data, thereby leading to poor ACC in general. Compared to the baselines, I-BAU exhibits the least sensitivity to the poison ratio, maintaining good ACC and ASR across different ratios.

| poison ratio | Results | No Def. | NC | DI | TABOR | FP | NAD | DP | Ours |
|---|---|---|---|---|---|---|---|---|---|
| 5.0% | **ACC** | 86.58 | 83.14 | 78.63 | × | 83.16 | 79.74 | 36.80 | 84.76 |
| | **ASR** | 99.88 | 5.58 | 10.40 | × | 99.72 | 6.34 | 96.84 | 9.78 |
| 0.5% | **ACC** | 86.42 | 84.16 | 83.56 | × | 84.72 | 80.92 | 39.92 | 83.22 |
| | **ASR** | 98.58 | 12.9 | 20.22 | × | 93.78 | 28.6 | 61.27 | 13.08 |

Table 4: Results on CIFAR-10 (Trojan WM) with different poison ratios. × marks no trigger was detected.

Table 5 shows sensitivity to available clean data's size. As DP is independent from the clean set (which trains a general robust model against all perturbations from scratch), we drop it from the comparison. We see that as the number of clean samples drops, the performance of all defenses declines. I-BAU is least sensitive to the size of clean samples; even in extreme case with only 100 clean samples available, I-BAU still maintains an acceptable performance of removing backdoors.

| # Clean Data | Results | No Def. | NC | DI | TABOR | FP | NAD | Ours |
|---|---|---|---|---|---|---|---|---|
| 2,500 | **ACC** | 84.92 | 78.39 | 80.63 | 80.23 | 81.36 | 46.8 | 82.21 |
| | **ASR** | 99.96 | 6.53 | 10.07 | 33.40 | 99.58 | 7.12 | 6.96 |
| 500 | **ACC** | 84.92 | 78.24 | 80.17 | 77.03 | 78.1 | 38.5 | 80.07 |
| | **ASR** | 99.96 | 25.66 | 1.14 | 21.92 | 85.68 | 9.08 | 5.20 |
| 100 | **ACC** | 84.92 | 84.101 | 69.51 | 83.495 | 73.00 | 36.14 | 76.9 |
| | **ASR** | 99.96 | 99.92 | 1.12 | 99.687 | 97.80 | 5.76 | 4.00 |

Table 5: Results with different # of clean data on CIFAR-10 (Trojan WM).

## 6.4 EFFICIENCY

Finally, we compare the efficiency of defenses. We use the one-trigger-one-target attack setting to exemplify the result. Table 6 shows the average runtime for each defense to take effect (i.e., mitigating the ASR to $< 20\%$). As I-BAU can mitigate the attacks in one iteration, it only takes 6.82 s on average on CIFAR-10 and 7.84 s on GTSRB. Note that NC and TABOR need to go through each label independently for trigger synthesis; thus, the total runtime is proportional to the number of labels. Especially, it takes much more time for the two to take effect on GTSRB than on CIFAR-10. In difficult attack cases, such as all-to-all attacks and

| | CIFAR-10 (s) | GTSRB (s) |
|---|---|---|
| NC | 384.92 | 1864.96 |
| DI | 394.38 | 472.21 |
| TABOR | 1123.31 | 3529.70 |
| FP | 45.33 | 83.78 |
| NAD | 79.90 | 79.14 |
| Ours | **6.82** | **7.84** |

Table 6: Average time for defenses to be effective on one-trigger-one-target cases.

multi-trigger-multi-target attacks, I-BAU requires more rounds to be operative but remains the **only** effective one across all settings with high efficiency and efficacy. The Appendix shows the performance of I-BAU over different rounds under multi-target attack cases. Theoretical analysis and comparison of the time complexity of I-BAU are presented in Appendix A.3.3.

## 7 CONCLUSION

In this work, we proposed a minimax formulation of the backdoor removal problem. This formulation encompassed the objective of the existing unlearning work without making assumptions about attack strategies or trigger patterns. To solve the proposed minimax, we proposed I-BAU using implicit hypergradients. A theoretical analysis verified the convergence and generalizability of the proposed I-BAU or minimax formulation. A comprehensive empirical study established that I-BAU is the only generalizable defense across all evaluated eleven backdoor attacks. The defense results are comparable to or exceed the best results obtained by combining six existing state-of-the-art techniques. Meanwhile, I-BAU is less sensitive to poison rate and is effective in extreme cases where the defender has access to only 100 clean samples. Finally, under the standard one-trigger-one-target circumstances, I-BAU can achieve an effective defense in an average of 7.35 s.

## ACKNOWLEDGEMENTS

This work was supported by the Commonwealth Cyber Initiative, an investment in the advancement of cyber R&D, innovation, and workforce development. For more information about CCI, visit www.cyberinitiative.org

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

# A  APPENDIX

## A.1  CONVERGENCE BOUND

Consider the minimax formulation of backdoor unlearning defined in (1). To simply the notation, we will use $\theta$ instead of $\theta_i$, unless otherwise specified. We start by defining $\Phi(\delta, \theta) = \delta + \alpha(\theta)\nabla_1 H(\delta, \theta)$, where $H$ (sum of cross-entropies) is twice continuously differentiable w.r.t. both $\delta$ and $\theta$. Recall that $H(\cdot, \theta)$ is $\mu_H(\theta)$-strongly convex and $L_H(\theta)$-Lipschitz smooth, where $\mu_H(\cdot)$ and $L_H(\cdot)$ are continuously differentiable. By setting the step size $\alpha(\theta) = \frac{2}{L_H(\theta)+\mu_H(\theta)}$, it can be shown that $\Phi(\cdot, \theta)$ is a contraction for some coefficient $q_\theta := \max\{1 - \alpha(\theta)\mu_H(\theta), \alpha(\theta)L_H(\theta) - 1\}$. The optimal choice of the step-size leads to $q_\theta = \frac{\kappa(\theta)-1}{\kappa(\theta)+1}$, where $\kappa(\theta) = \frac{L_H(\theta)}{\mu_H(\theta)}$. Note that, for every $i \in (1, K), \theta \in \Theta$,

$$\mu_H(\theta)\mathbb{I} \preccurlyeq \nabla_1^2 H\left(\delta_i(\theta), \theta\right) \preccurlyeq L_H(\theta)\mathbb{I}. \tag{7}$$

Hence, the condition number of $\nabla_1^2 H\left(\delta_i(\theta), \theta\right)$ is smaller than $\kappa(\theta)$ (Grazzi et al., 2020).

Let $\partial_1 \Phi(\delta, \theta)$ and $\partial_2 \Phi(\delta, \theta)$ denote the partial Jacobians of $\Phi$ at $(\delta, \theta)$ w.r.t. the first and the second variables, respectively. We can write the derivatives of $\Phi$ as:

$$\begin{aligned} \partial_2\Phi(\delta,\theta) &= -\nabla_1 H(\delta,\theta)\nabla\alpha(\theta)^\top - \alpha(\theta)\nabla_{21}^2 H(\delta,\theta), \\ \partial_1\Phi(\delta,\theta) &= I - \alpha(\theta)\nabla_1^2 H(\delta,\theta). \end{aligned} \tag{8}$$

When evaluated at $(\delta(\theta), \theta)$, the partial Jacobian w.r.t. the second variable, $\partial_2 \Phi(\delta(\theta), \theta)$, can be simplified as:

$$\partial_2\Phi(\delta,\theta) = -\alpha(\theta)\nabla_{21}^2 H(\delta(\theta),\theta). \tag{9}$$

Thus, $\|\partial_2\Phi(\delta(\theta),\theta)\| = \alpha(\theta)\|\nabla_{21}^2 H(\delta(\theta),\theta)\|$. Following the notations from (Grazzi et al., 2020), we create the bound of the partial Jacobian of $\Phi$ to be $L_{\Phi,\theta} \geq \|\partial_2\Phi(\delta(\theta),\theta)\|$. By our assumption, the bound for the cross derivative term $\delta$ is $L_{h,\theta} \geq \|\nabla_{21}^2 H(\delta(\theta),\theta)\|$. Hence, we can choose:

$$L_{\Phi,\theta} = \alpha(\theta)L_{h,\theta}. \tag{10}$$

Let $\Delta_{\partial_1\Phi} := \|\partial_1\Phi\left(\delta_1, \theta\right) - \partial_1\Phi\left(\delta_2, \theta\right)\|$ and $\Delta_{\partial_2\Phi} := \|\partial_2\Phi\left(\delta_1, \theta\right) - \partial_2\Phi\left(\delta_2, \theta\right)\|$. By assumption on the *Lipschitz continuity and norm boundedness of second-order terms*, we have :

$$\Delta_{\partial_1\Phi} = \left\|\alpha(\theta)\left(\nabla_1^2 H\left(\delta_1, \theta\right) - \nabla_1^2 H\left(\delta_2, \theta\right)\right)\right\| \leq \alpha(\theta)\hat{\rho}_{1,\theta}\left\|\delta_1 - \delta_2\right\|, \tag{11}$$

and:

$$\begin{aligned} \Delta_{\partial_2\Phi} =& \left\|\left(\nabla_1 H\left(\delta_1, \theta\right) - \nabla_1 H\left(\delta_2, \theta\right)\right)\nabla\alpha(\theta)^\top + \alpha(\theta)\left(\nabla_{21}^2 H\left(\delta_1, \theta\right) - \nabla_{21}^2 H\left(\delta_2, \theta\right)\right)\right\| \\ &\leq \left(L_H(\theta)\|\nabla\alpha(\theta)\| + \alpha(\theta)\hat{\rho}_{2,\theta}\right)\left\|\delta_1 - \delta_2\right\|. \end{aligned} \tag{12}$$

Thus, $\partial_1\Phi(\cdot, \theta)$ is Lipschitz continuous with constant

$$v_{1,\theta} := \alpha(\theta)\hat{\rho}_{1,\theta}, \tag{13}$$

and $\partial_2\Phi(\cdot, \theta)$ is Lipschitz continuous with constant

$$v_{2,\theta} := L_H(\theta)\|\nabla\alpha(\theta)\| + \alpha(\theta)\hat{\rho}_{2,\theta}. \tag{14}$$

Recall the result from (Grazzi et al., 2020):

**Lemma 1** *The approximate hypergradient has an error bounded by*

$$\left\|\nabla\bar{\psi}(\theta) - \nabla\psi^*(\theta)\right\| \leq \left(c_1(\theta) + c_2(\theta)\frac{1 - q_\theta^t}{1 - q_\theta}\right)\rho_\theta(t) + c_3(\theta)q_\theta^t, \tag{15}$$

*where*

$$\begin{aligned} c_1(\theta) &= \left(\eta_{2,\theta} + \frac{\eta_{1,\theta}L_{\Phi,\theta}}{1 - q_\theta}\right)C_\delta, \\ c_2(\theta) &= \left(\nu_{2,\theta} + \frac{\nu_{1,\theta}L_{\Phi,\theta}}{1 - q_\theta}\right)L_{H,\theta}C_\delta, \\ c_3(\theta) &= \frac{L_{H,\theta}L_{\Phi,\theta}}{1 - q_\theta}. \end{aligned} \tag{16}$$

The proof of Theorem 1 uses the inequality from Lemma 1 with the constants specified in (10), (13), and (14).

## A.2 GENERALIZATION BOUNDS

Following the notations from (Bartlett et al., 2017), we define a general margin operator $\mathcal{M}(v, y) := v_y - \max_{j \neq y} v_j$, and the ramp loss $\ell_\gamma$ with a given margin $\gamma$:

$$\ell_\gamma(r) := \begin{cases} 0 & r < -\gamma \\ 1 + r/\gamma & r \in [-\gamma, 0] \\ 1 & r > 0 \end{cases} . \tag{17}$$

According to (Yin et al., 2019), the population risk against a perturbation $\delta$ is given by:

$$R_\gamma(\theta) := \mathbb{E}\left(\ell_\gamma(-\mathcal{M}(\theta(x + \delta), y))\right), \tag{18}$$

and the empirical risk is given by:

$$\hat{R}_\gamma(\theta) := n^{-1} \sum_j \ell_\gamma\left(-\mathcal{M}\left(\theta\left(x_j + \delta\right), y_j\right)\right). \tag{19}$$

Note that $R_\gamma(\theta)$ and $\hat{R}_\gamma(\theta)$ are the upper bounds of the fraction of errors on the source distribution and the dataset used for unlearning, $D_{san}$, respectively. Finally, given a set of real-valued functions $\mathcal{H}$, recall the *Rademacher complexity* as:

$$\Re\left(\mathcal{H}_{|S}\right) := n^{-1} \mathbb{E}_\varsigma \sup_{h \in \mathcal{H}} \sum_{j=1}^n \varsigma_j h\left(x_j, y_j\right), \tag{20}$$

where $\varsigma_j \in \{\pm 1\}$ is the Rademacher random variable. The following bound of the adversarial unlearning can be derived using standard tools in Rademacher complexity.

**Lemma 2** *Given unlearned models $\Theta$ with $\theta \in \Theta$ and some margin $\gamma > 0$, define:*

$$\Theta_\gamma := \{(x, y) \mapsto \ell_\gamma(-\mathcal{M}(\theta(x + \delta), y)) : \theta \in \Theta\}. \tag{21}$$

*The following holds with the probability of at least $1 - \xi$ over the clean dataset, $D_{san}$ (see Algorithm 1) for every $\theta \in \Theta$:*

$$\Pr\left[\arg\max_j [\theta(x + \delta)_j] \neq y\right] \leq \hat{R}_\gamma(\theta) + 2\Re\left((\Theta_\gamma)_{|D_{san}}\right) + 3\sqrt{\frac{\ln(1/\xi)}{2n}}. \tag{22}$$

To instantiate this bound, we only need to control the Rademacher complexity, $\Re\left((\Theta_\gamma)_{|D_{san}}\right)$, for linear models and neural networks.

### A.2.1 PROOF FOR LINEAR MODELS

Define the set of linear, poisoned models under perturbation $\delta$:

$$\Theta_\gamma^{lin} := \{(x, y) \mapsto \langle \theta, x + \delta \rangle : \theta \in \Theta\}, \tag{23}$$

where $\theta$ can be regarded as a weight matrix $\|\theta\|$ bounded by $C_\theta$. Recall that the perturbation norm $\|\delta\|$ is bounded by $C_\delta$, and the input $l2$ norm $\|x\|_2$ is bounded by $\chi$. We can proceed to evaluate the

upper bound of the Rademacher complexity:

$$
\begin{aligned}
\Re\left((\Theta_\gamma)_{|D_{san}}\right) &= \mathbb{E}_\varsigma\left[\sup_{\|\theta\|\le C_\theta} \frac{1}{n}\sum_{j=1}^n \varsigma_j\langle\theta, x+\delta\rangle\right] \\
&\le \mathbb{E}_\varsigma\left[\sup_{\|\theta\|\le C_\theta} \frac{1}{n}\left\|\sum_{j=1}^n \varsigma_j(x_j+\delta)\right\|\,\|\theta\|\right] \quad \text{(Cauchy-Schwarz inequality)} \\
&\le C_\theta\mathbb{E}_\varsigma\left[\frac{1}{n}\left\|\sum_{j=1}^n \varsigma_j(x_j+\delta)\right\|\right] \quad \text{(given that } \|\theta\|\le C_\theta) \\
&= C_\theta\mathbb{E}_\varsigma\left[\frac{1}{n}\left\|\sum_{j=1}^n \varsigma_j x_j + \delta\sum_{j=1}^n \varsigma_j\right\|\right] \\
&\le C_\theta\mathbb{E}_\varsigma\left[\frac{1}{n}\left\|\sum_{j=1}^n \varsigma_j x_j\right\| + \frac{1}{n}\left\|\delta\sum_{j=1}^n \varsigma_j\right\|\right] \quad \text{(Triangle inequality)} \\
&= C_\theta\left(\underbrace{\mathbb{E}_\varsigma\frac{1}{n}\left\|\sum_{j=1}^n \varsigma_j x_j\right\|}_{\mathcal{A}} + \underbrace{\mathbb{E}_\varsigma\frac{1}{n}\left\|\delta\sum_{j=1}^n \varsigma_j\right\|}_{\mathcal{B}}\right).
\end{aligned}
\tag{24}
$$

In particular, $\mathcal{A}$ can be bounded by:

$$
\begin{aligned}
\mathcal{A} &= \mathbb{E}_\varsigma\frac{1}{n}\sqrt{\sum_{j=1}^n\sum_{k=1}^n \varsigma_j\varsigma_k\langle x_j, x_k\rangle} \\
&\le \frac{1}{n}\sqrt{\mathbb{E}_\varsigma\sum_{j,k}^n \varsigma_j\varsigma_k\langle x_j, x_k\rangle} \quad \text{(Jensen's inequality)} \\
&\le \frac{1}{n}\sqrt{\sum_{j=1}^n \|x_i\|^2} \\
&\le \frac{\chi}{\sqrt{n}}.
\end{aligned}
\tag{25}
$$

Similarly, $\mathcal{B}$ can be bounded by:

$$
\begin{aligned}
\mathcal{B} &\le \mathbb{E}_\varsigma\sup_{\|\delta\|\le C_\delta} \frac{1}{n}\left\|\delta\sum_{j=1}^n \varsigma_j\right\| \\
&\le \frac{C_\delta}{n}\mathbb{E}_\varsigma\left\|\sum_{j=1}^n \varsigma_j\right\| \quad \text{(given that } \|\delta\|\le C_\delta) \\
&\le \frac{C_\delta}{n}\sqrt{\mathbb{E}_\varsigma\sum_{j,k}^n \varsigma_j\varsigma_k} \quad \text{(Jensen's inequality)} \\
&= \frac{C_\delta}{\sqrt{n}}.
\end{aligned}
\tag{26}
$$

Thus, we can bound the Rademacher complexity as following:

$$
\Re\left((\Theta_\gamma)_{|D_{san}}\right) \le \frac{C_\theta(\chi+C_\delta)}{\sqrt{n}}.
\tag{27}
$$

Substituting (27) to the (22) completes the proof of Theorem 2.

A.2.2 PROOF FOR NEURAL NETWORKS

To instantiate the bound with Rademacher complexity for neural networks, we will follow the idea from (Bartlett et al., 2017) and use covering numbers to bound the Rademacher complexity, $\Re\left((\Theta_\gamma)_{|D_{san}}\right)$. Let $\mathcal{N}(U, \epsilon, \|\cdot\|)$ denotes the least cardinality of any subset $V \subseteq U$ that covers U at a scale $\epsilon$ with norm $\|\cdot\|$, i.e., $\sup_{A \in U} \min_{B \in V} \|A - B\| \leq \epsilon$. Recall the Dudley entropy integral below.

**Lemma 3** *Assume that the input values are norm bounded by* 1,

$$\Re\left((\Theta_\gamma)_{|D_{san}}\right) \leq \inf_{\alpha > 0}\left(\frac{4\alpha}{\sqrt{n}} + \frac{12}{n}\int_\alpha^{\sqrt{n}}\sqrt{\log \mathcal{N}\left((\Theta_\gamma)_{|D_{san}}, \epsilon, \|\cdot\|_2\right)}d\epsilon\right). \tag{28}$$

Thus, the derivation of the generalization bound of (1) with neural networks is reduced to the problem of finding the bound for the covering number of the set of all neural networks, $\mathcal{N}\left((\Theta_\gamma)_{|D_{san}}, \epsilon, \|\cdot\|_2\right)$. The full problem can be divided into four steps: **(I)** finding a matrix-covering bound for the affine transformation of the input layer of poisoned models, conducted in this section; **(II)** finding a matrix-covering bound for the affine transformation of the following layers after the first layer, provided in (Bartlett et al., 2017); **(III)** obtaining a covering number bound for entire networks using induction on layers; **(IV)** accomplishing the complete generalization bound for neural networks by applying the covering number to (22) and (28).

**Step (I)**: **Input Layer Matrix Covering**. The covering number of the input layer considers the matrix product, $\hat{Z} = (X + \Delta)A_1$, where $A_1 \in \mathbb{R}^{d \times m}$ is the weight matrix of the input layer, $X \in \mathbb{R}^{n \times d}$ is the input data, and $\Delta = \underbrace{[\delta, \delta, \ldots, \delta]}_{n}^\top$ is the UNO perturbation vector repeated $n$ times to get the same shape as $X$. Note that $\hat{Z}$ can be rewritten as:

$$\begin{aligned}
\hat{Z} &= (X + \Delta)A_1 \\
&= \begin{bmatrix} X & \mathbb{I}_n \end{bmatrix}\begin{bmatrix} A_1 \\ \Delta A_1 \end{bmatrix} \\
&= \hat{X}\hat{A}_1,
\end{aligned} \tag{29}$$

where $\mathbb{I}_n$ is the $n \times n$ identity matrix, $\hat{X}$ is an $n \times (d + n)$ matrix, and $\hat{A}_1$ is a $(d + n) \times m$ matrix. Now, the goal of step **(I)** is to find the covering number for the matrix product, $\hat{X}\hat{A}_1$.

Given $X \in \mathbb{R}^{n \times d}$, we can obtain the normalized matrix $Y \in \mathbb{R}^{n \times d}$ by rescaling the columns of $X$ to have unit $p$-norm: $Y_{:,j} := X_{:,j}/\|X_{:,j}\|_p$. Let $N_0 := 2(n + d)m$, and define

$$\left\{\hat{V}_1, \ldots, \hat{V}_{N_0}\right\} := \left\{g\begin{bmatrix} Y & \mathbb{I}_n \end{bmatrix}\mathbf{e}_i\mathbf{e}_j^\top : g \in \{-1, +1\}, i \in \{1, \ldots, d + n\}, j \in \{1, \ldots, m\}\right\}. \tag{30}$$

For $p \leq 2$, the results from (Bartlett et al., 2017) implies:

$$\max_i\left\|\hat{V}_i\right\|_2 \leq \max_{i \in \{1 \ldots N_0\}}\left\|\begin{bmatrix} Y & \mathbb{I}_n \end{bmatrix}\mathbf{e}_i\right\|_2 = \max_i\frac{\left\|\hat{X}\mathbf{e}_i\right\|_2}{\left\|\hat{X}\mathbf{e}_i\right\|_p} \leq 1. \tag{31}$$

Define $\alpha_0 \in \mathbb{R}^{(d+n) \times m}$ to be a "rescaling matrix":

$$\alpha_0 = \begin{bmatrix}
\|X_{:,1}\|_p & \cdots & \|X_{:,1}\|_p \\
\|X_{:,2}\|_p & \cdots & \|X_{:,2}\|_p \\
\vdots & \ddots & \vdots \\
\|X_{:,d}\|_p & \cdots & \|X_{:,d}\|_p \\
1 & \cdots & 1 \\
\vdots & \ddots & \vdots \\
1 & \cdots & 1
\end{bmatrix}, \tag{32}$$

where the purpose of $\alpha_0$ is to annul the rescaling of $\hat{X}$ introduced by $\hat{Y} = [\ Y\ \ \mathbb{I}_n\ ]$, i.e., $\hat{X}\hat{A}_1 = \hat{Y}(\alpha_0 \odot \hat{A}_1)$, and $\odot$ denotes the element-wise product. Let $\hat{B} := \alpha_0 \odot \hat{A}_1$. Then, we have:

$$
\begin{aligned}
(X + \Delta)A_1 &= \hat{X}\hat{A}_1 \\
&= \hat{Y}\hat{B} \\
&= \hat{Y} \sum_{i=1}^{n+d} \sum_{j=1}^{m} \hat{B}_{ij} e_i e_j^{\top} \\
&= \hat{Y}\|\hat{B}\|_1 \sum_{i=1}^{n+d} \sum_{j=1}^{m} \frac{\hat{B}_{ij}}{\|\hat{B}\|_1} e_i e_j^{\top} \\
&= \|\hat{B}\|_1 \sum_{i=1}^{n+d} \sum_{j=1}^{m} \frac{\hat{B}_{ij}}{\|\hat{B}\|_1} [\ Y\ \ \mathbb{I}_n\ ] e_i e_j^{\top} \\
&\in \|\hat{B}\|_1 \cdot \mathrm{conv}\left(\left\{\hat{V}_1, \ldots, \hat{V}_{N_0}\right\}\right),
\end{aligned}
\tag{33}
$$

where $\mathrm{conv}\left(\left\{\hat{V}_1, \ldots, \hat{V}_{N_0}\right\}\right)$ is the convex hull of $\left\{\hat{V}_1, \ldots, \hat{V}_{N_0}\right\}$. Given conjugate exponents $(p, q)$ and $(r, s)$ with $p \leq 2$, by the conjugacy of $\|\cdot\|_{p,r}$ and $\|\cdot\|_{q,s}$:

$$
\|\hat{B}\|_1 \leq \langle \alpha_0, |\hat{A}_1| \rangle \leq \|\alpha_0\|_{p,r} \|\hat{A}_1\|_{q,s}.
\tag{34}
$$

Defining $\mathcal{C}$ as the desired cover of the first layer $\hat{X}\hat{A}_1$:

$$
\mathcal{C} := \left\{ \frac{\|\hat{B}\|_1}{k} \sum_{i=1}^{N_0} k_i V_i : k_i \geq 0, \sum_{i=1}^{N_0} k_i = k \right\} = \left\{ \frac{\|\hat{B}\|_1}{k} \sum_{j=1}^{k} \hat{V}_{i_j} : (i_1, \ldots, i_k) \in [N_0]^k \right\},
\tag{35}
$$

where, $k := \left\lceil \frac{a_0{}^2 (1 + mn^{\frac{1}{2}} C_\delta)^2 (\|X\|_p + n^{\frac{1}{2}})^2 m^{\frac{2}{r}}}{\epsilon_1{}^2} \right\rceil$. By construction, $|\mathcal{C}| \leq [N_0]^k$. Now, we prove that $\mathcal{C}$ is the desired cover set. The technique is generalized from (Bartlett et al., 2017) to backdoor unlearning.

Consider the case of $\|A_1\|_{2,1}$, i.e., $(q, s) = (2, 1)$, and let $\|A_1\|_{2,1} \leq a_0$, then we get:

$$
\begin{aligned}
\|\alpha_0\|_{p,r} &= \left\| \left( \|\alpha_{:,1}\|_p, \ldots, \|\alpha_{:,m}\|_p \right) \right\|_r \\
&= \left\| \left( \left\| \left( \|X_{:,1}\|_p, \ldots, \|X_{i,d}\|_p, 1, \ldots, 1 \right) \right\|_p, \ldots, \left\| \left( \|X_{:,1}\|_p, \ldots, \|X_{:,d}\|_p, 1, \ldots, 1 \right) \right\|_p \right) \right\|_r \\
&= m^{1/r} \left\| \left( \|X_{:,1}\|_p, \ldots, \|X_{:,d}\|_p, \overbrace{1, \ldots, 1}^{n} \right) \right\|_p \\
&= m^{1/r} \left( \sum_{j=1}^{d} \|X_{:,j}\|_p^p + n \right)^{1/p} \\
&\leq m^{1/r} \left( \|X\|_p + n^{1/p} \right).
\end{aligned}
\tag{36}
$$

Subsequently, the bound of $\left\|\hat{A}_1\right\|_{2,1}$ can be derived from:

$$
\begin{aligned}
\left\|\hat{A}_1\right\|_{2,1} &= \left\| \begin{bmatrix} A_1 \\ \Delta A_1 \end{bmatrix} \right\|_{2,1} \\
&= \left\| \left( \left\| \begin{bmatrix} A_{:1} \\ \Delta A_{:1} \end{bmatrix} \right\|_2, \ldots, \left\| \begin{bmatrix} A_{:m} \\ \Delta A_{:m} \end{bmatrix} \right\|_2 \right) \right\|_1 \\
&= \left\| \left( \left( \|A_{:1}\|_2^2 + \|\Delta A_{:1}\|_2^2 \right)^{1/2}, \ldots, \left( \|A_{:m}\|_2^2 + \|\Delta A_{:m}\|_2^2 \right)^{1/2} \right) \right\|_1,
\end{aligned}
\tag{37}
$$

where we have:

$$
\begin{aligned}
\|\Delta A_{:i}\|_2^2 &= \left\| \begin{bmatrix} \delta^\top \\ \vdots \\ \delta^\top \end{bmatrix} A_{:i} \right\|_2^2 \\
&= \left\| \begin{bmatrix} \delta^\top A_{:i} \\ \vdots \\ \delta^\top A_{:i} \end{bmatrix} \right\|_2^2 \\
&= \left( \delta^\top A_{:i} \right)^2 + \ldots + \left( \delta^\top A_{:i} \right)^2 \\
&= n \left( \delta^\top A_{:i} \right)^2 .
\end{aligned}
\tag{38}
$$

Therefore, substituting (38) to (37) gives us the bound:

$$
\begin{aligned}
\left\| \hat{A}_1 \right\|_{2,1} &= \left\| \left( \|A_{:1}\|_2^2 + n \left( \delta^\top A_{:1} \right)^2 \right)^{1/2}, \ldots, \left( \|A_{:m}\|_2^2 + n \left( \delta^\top A_{:m} \right)^2 \right)^{1/2} \right\|_1 \\
&\leq \left\| \|A_{:1}\|_2 + n^{1/2} \left\| \delta^\top A_{:1} \right\|, \ldots, \|A_{:m}\|_2 + n^{1/2} \left\| \delta^\top A_{:m} \right\| \right\|_1 \\
&= \|A_{:1}\|_2 + n^{1/2} \left\| \delta^\top A_{:1} \right\| + \ldots + \|A_{:m}\|_2 + n^{1/2} \left\| \delta^\top A_{:m} \right\| \\
&= \|A_1\|_{2,1} + n^{1/2} \sum_{j=1}^m \left\| \delta^\top A_{:j} \right\| \\
&\leq \|A_1\|_{2,1} + n^{1/2} \sum_{j=1}^m \left( \|\delta\|_2 \cdot \|A_{:j}\|_2 \right) \\
&= \|A_1\|_{2,1} \left( 1 + mn^{1/2}\|\delta\|_2 \right) \\
&\leq a_0 \left( 1 + mn^{1/2}C_\delta \right) .
\end{aligned}
\tag{39}
$$

Let $a_1 = a_0 \left( 1 + mn^{1/2}C_\delta \right)$. Given the case of $\|A_1\|_{2,1}$, we can obtain the bound for $\|\hat{B}\|_1$:

$$
\|\hat{B}\|_1 \leq \|\alpha_0\|_{p,r} \|\hat{A}_1\|_{2,1} \leq m^{1/r} \left( \|X\|_p + n^{1/p} \right) a_1 .
\tag{40}
$$

Thus, combining (31) and following the Maurey lemma (Pisier (1981), Zhang (2002), Lemma 1), we have:

$$
\begin{aligned}
\left\| (X + \Delta)A_1 - \frac{\|\hat{B}\|_1}{k} \sum_{i=1}^{N_0} k_i \hat{V}_i \right\|_2^2 &\leq \frac{\|\hat{B}\|_1^2}{k} \max_{i=1 \ldots N_0} \left\| \hat{V}_i \right\|_2^2 \\
&\leq \frac{a_1^2 (\|X\|_p + n^{\frac{1}{2}})^2 m^{\frac{2}{r}}}{k} \\
&\leq \epsilon_1^2,
\end{aligned}
\tag{41}
$$

which shows that the desired cover element is in $\mathcal{C}$.

**Theorem 4** *In the case of the input layer of a poisoned model, $\theta$, consider $\Delta$ as the UNO perturbation matrix with $n$ rows of $\delta$, and $A_1 \in \mathbb{R}^{d \times m}$ to be the weight metrix of the first layer. The covering resolution $\epsilon_1$ is given. Defining $a_1 := a_0 \left( 1 + mn^{1/2}C_\delta \right)$, where $a_0$ is the bound of $\|A_1\|_{2,1}$, for any input $X \in \mathbb{R}^{n \times d}$, the convering number is bounded as follows:*

$$
\ln \mathcal{N} \left( \left\{ (X + \Delta)A_1 : A_1 \in \mathbb{R}^{d \times m} \right\}, \epsilon_1, \|\cdot\|_2 \right) \leq \left\lceil \frac{a_1^2 (\|X\|_p + n^{1/2})^2 m^{2/r}}{\epsilon_1^2} \right\rceil \ln(2(d+n)m).
\tag{42}
$$

**Step (II)**: **Other Layer's Matrix Covering**. Let's define $Z \in \mathbb{R}^{n_i \times d_i}$ as the input of a specific layer of the neural network, and $A_i \in \mathbb{R}^{d_i \times m_i}$ as the weight matrix of that layer. Given conjugate exponents $(p, q)$ and $(r, s)$ with $p \leq 2$, let$\|A_i\|_{q,s} \leq a_i$ . Lastly, the covering resolution of each

layer, $\epsilon_i$, is given. Since no perturbation is considered in the following layers, we can directly adopt the covering studied in (Bartlett et al., 2017) as follows:

$$\ln \mathcal{N}\left(\left\{ZA_i : A_i \in \mathbb{R}^{d_i \times m_i}, \|A_i\|_{q,s} \leq a_i\right\}, \epsilon_i, \|\cdot\|_2\right) \leq \left\lceil \frac{a_i^2 \|Z\|_p^2 m_i^{2/r}}{\epsilon_i^2} \right\rceil \ln(2d_i m_i). \quad (43)$$

**Step (III)**: **The Whole Network Covering Bound**. Recall that the whole neural network is structured as follows: $F_\theta(x) := \sigma_L\left(A_L \sigma_{L-1}\left(A_{L-1} \ldots \sigma_1\left(A_1 x\right) \ldots\right)\right)$, where $\sigma_i$ is $\varrho_i$-Lipschitz.

Define two sequences of vector spaces $\mathcal{V}_1, \ldots, \mathcal{V}_L$ and $\mathcal{W}_2, \ldots, \mathcal{W}_{L+1}$, where $\mathcal{V}_i$ has a norm $|\cdot|_i$ and $\mathcal{W}_i$ has a norm $\||\cdot\||_i$. The linear operators, $A_i : \mathcal{V}_i \to \mathcal{W}_{i+1}$, are associated with some operator norm $|A_i|_{i \to i+1} \leq s_i$, i.e., $|A_i|_{i \to i+1} := \|A_i\|_\sigma \leq \sup_{|Z|_i \leq 1} \||A_i Z\||_{i+1} = s_i$. Then, letting $\tau := \sum_{j \leq L} \epsilon_j \varrho_j \prod_{l=j+1}^L \varrho_l s_l$, with given convering resolutions, $(\epsilon_1, \ldots, \epsilon_L)$, the neural net images, $\mathcal{H}_X := \{F_\theta(X + \Delta)\}$, have the covering number bound (Bartlett et al., 2017):

$$\mathcal{N}\left(\mathcal{H}_X, \tau, |\cdot|_{L+1}\right) \leq \prod_{i=1}^L \sup_{\substack{(A_1, \ldots, A_{i-1}) \\ \forall j < i}} \mathcal{N}\left(\left\{A_i F_{(A_1, \ldots, A_{i-1})}(X + \Delta)\right\}, \epsilon_i, \||\cdot\||_{i+1}\right). \quad (44)$$

**Step (IV)**: **Proof of Theorem 3**. The key technique in the remainder of this proof is

- **1)** to substitute covering number estimates from (42) and (43) into (44) but
- **2)** centering the covers at 0 (meaning the cover at layer $i \in (2, L)$ satisfies $\|A_i\|_{2,1} \leq a_i$,

  $\|A_1\|_{2,1} \leq a_0$, and $\|\hat{A}_1\|_{2,1} = \left\|\begin{bmatrix} A_1 \\ \Delta A_1 \end{bmatrix}\right\|_{2,1} \leq a_1$), and

- **3)** collecting $(x_1, \ldots, x_n)$ as rows of matrix $X \in \mathbb{R}^{n \times d}$.

To start, the covering number estimate of the whole network from (44) when combined with (42) and (43) (specifically with $p = 2, s = 1$, and $W = \max(d_0, \ldots, d_L)$) results in:

$$
\begin{aligned}
&\ln \mathcal{N}\left(\mathcal{H}_X, \epsilon, \|\cdot\|_2\right) \\
&\leq \sum_{i=1}^L \sup_{\substack{(\hat{A}_1, A_2, \ldots, A_{i-1}) \\ \forall j < i}} \ln \mathcal{N}\left(\left\{A_i F_{(\hat{A}_1, A_2 \ldots, A_{i-1})}\left(\begin{bmatrix} X^\top \\ \mathbb{I}_n \end{bmatrix}\right)\right\}, \epsilon_i, \|\cdot\|_2\right) \\
&\overset{(*)}{=} \sup_{A_1} \ln \mathcal{N}\left(\left\{\hat{A}_1\left(\begin{bmatrix} X^\top \\ \mathbb{I}_n \end{bmatrix}\right)^\top\right\}, \epsilon_1, \|\cdot\|_2\right) \\
&\quad + \sum_{i=2}^L \sup_{\substack{(A_2, \ldots, A_{i-1}) \\ \forall j < i}} \ln \mathcal{N}\left(\left\{A_i F_{(\hat{A}_1, \ldots, A_{i-1})}\left(\begin{bmatrix} X^\top \\ \mathbb{I}_n \end{bmatrix}^\top\right)\right\}, \epsilon_i, \|\cdot\|_2\right) \\
&\leq \sup_{\hat{A}_1} \frac{a_1^2 \left\|\hat{A}_1\left(\begin{bmatrix} X^\top \\ \mathbb{I}_n \end{bmatrix}\right)^\top\right\|_2^2}{\epsilon_1^2} \ln\left(2W(W+n)\right) \\
&\quad + \sum_{i=2}^L \sup_{\substack{(A_2, \ldots, A_{i-1}) \\ \forall j < i}} \frac{a_i^2 \left\|F_{(\hat{A}_1, A_2, \ldots, A_{i-1})}\left(\begin{bmatrix} X^\top \\ \mathbb{I}_n \end{bmatrix}\right)^\top\right\|_2^2}{\epsilon_i^2} \ln\left(2W^2\right),
\end{aligned} \quad (45)
$$

where $(*)$ equality holds since **1)** $l_2$ coverings of a matrix and its transpose are the same, and **2)** the cover can be translated by $F_{(\hat{A}_1, A_2, \ldots, A_{i-1})}\left(\begin{bmatrix} X^\top \\ \mathbb{I}_n \end{bmatrix}\right)^\top A_i^\top$ without changing its cardinality. We can further simplify (45), by evaluating the following norm for any $A_i \in (A_2, \ldots, A_{L-1})$:

$$
\begin{aligned}
\left\|F_{(\hat{A}_1, A_2, \ldots, A_{i-1})}\left(\begin{bmatrix} X^\top \\ \mathbb{I}_n \end{bmatrix}\right)^\top\right\|_2 &= \left\|\sigma_{i-1}\left(A_{i-1} F_{(\hat{A}_1, A_2, \ldots, A_{i-1})}\left(\begin{bmatrix} X^\top \\ \mathbb{I}_n \end{bmatrix}\right)\right)\right\|_2 \\
&\leq \varrho_{i-1} \left\|A_{i-1} F_{(\hat{A}_1, A_2, \ldots, A_{i-1})}\left(\begin{bmatrix} X^\top \\ \mathbb{I}_n \end{bmatrix}\right)\right\|_2 \\
&\leq \varrho_{i-1} s_{i-1} \left\|F_{(\hat{A}_1, A_2, \ldots, A_{i-1})}\left(\begin{bmatrix} X^\top \\ \mathbb{I}_n \end{bmatrix}\right)\right\|_2,
\end{aligned} \quad (46)
$$

which by induction gives

$$\max_j \left\| F_{\left(\hat{A}_1, A_2, \ldots, A_{i-1}\right)} \left( \begin{bmatrix} X^\top \\ \mathbb{I}_n \end{bmatrix} \right)^\top \mathbf{e}_j \right\|_2 \leq (\|X\|_p + n^{1/2}) \prod_{j=1}^{i-1} \varrho_j s_j. \tag{47}$$

Combining (46) and (47), the cover is bounded by:

$$\begin{aligned}
&\ln \mathcal{N}\left(\mathcal{H}_X, \epsilon, \|\cdot\|_2\right) \\
&\leq \frac{a_1^2(\|X\|_p + \sqrt{n})^2}{\epsilon_1^2} \ln(2W(W+n)) + \sum_{i=2}^L \frac{a_i^2(\|X\|_p + \sqrt{n})^2 \varrho_1^2 s_1^2 \prod_{2 < j < i} \varrho_j^2 s_j^2}{\epsilon_i^2} \ln(2W^2) \\
&\leq \sum_{i=1}^L \frac{a_i^2(\|X\|_p + \sqrt{n})^2 \prod_{j<i} \varrho_j^2 s_j^2}{\epsilon_i^2} \ln(2W(W+n)).
\end{aligned} \tag{48}$$

Let

$$\epsilon_i := \frac{\alpha_i \epsilon}{\varrho_i \prod_{j>i} \varrho_j s_j} \quad \text{, where} \quad \alpha_i := \frac{1}{\bar{\alpha}} \left( \frac{a_i}{s_i} \right)^{2/3}, \quad \bar{\alpha} := \sum_{j=1}^L \left( \frac{a_j}{s_j} \right)^{2/3}, \tag{49}$$

then,

$$\ln \mathcal{N}\left(\mathcal{H}_X, \epsilon, \|\cdot\|_2\right) \leq \frac{(\|X\|_p + \sqrt{n})^2 \ln(2W(W+n)) \prod_{j=1}^L \varrho_j^2 s_j^2}{\epsilon^2} \left(\bar{\alpha}^3\right). \tag{50}$$

Consider the class of networks, $\Theta_\gamma^{NN}$, obtained by affixing the ramp loss, $\ell_\gamma$, and the negated margin operator, $-\mathcal{M}$, to the output of the provided network class:

$$\Theta_\gamma^{NN} := \{(x, y) \mapsto \ell_\gamma(-\mathcal{M}(\theta(x), y)) : \theta \in \Theta\}. \tag{51}$$

Since $(z, y) \mapsto \ell_\gamma(-\mathcal{M}(z, y))$ is $2/\gamma$-Lipschitz w.r.t. $\|\cdot\|_2$ and definition of $\ell_\gamma$, the function class $\Theta_\gamma^{NN}$ falls under the setting of (50), the covering number of a set of all neural networks is bounded as follows:

$$\begin{aligned}
&\ln \mathcal{N}\left(\left(\Theta_\gamma^{NN}\right)_{|x}, \epsilon, \|\cdot\|_2\right) \\
&\leq \frac{(\|X\|_p + \sqrt{n})^2 \ln(2W(W+n))}{\epsilon^2} \frac{4\left(\prod_{j=1}^L s_j^2 \varrho_j^2\right)\left(\sum_{i=1}^L \left(\frac{b_i}{s_i}\right)^{2/3}\right)^3}{\gamma^2} =: \frac{(\|X\|_p + \sqrt{n})^2 \ln(2W(W+n)) R}{\epsilon^2}.
\end{aligned} \tag{52}$$

Using the above covering number bound (52) in the Dudley entropy integral (28) with $\alpha := 1/\sqrt{n}$, we achieve the bound for the Rademacher complexity as follows:

$$\begin{aligned}
\mathfrak{R}\left((\mathcal{F}_\gamma)_{|S}\right) &\leq \inf_{\alpha>0} \left( \frac{4\alpha}{\sqrt{n}} + \ln(\sqrt{n}/\alpha) \frac{12\sqrt{(\|X\|_p + \sqrt{n})^2 \ln(2W(W+n)) R}}{n} \right) \\
&\leq \frac{4}{n} + \frac{24\ln(n)(\|X\|_p + \sqrt{n})\sqrt{\ln(2W(W+n))}}{\gamma n} \left( \prod_{i=1}^L s_i \varrho_i \right) \left( \sum_{i=1}^L \frac{a_i^{2/3}}{s_i^{2/3}} \right)^{3/2}.
\end{aligned} \tag{53}$$

We complete the proof of Theorem 3 by substituting (53) in (22).

### A.2.3 Emperical validation of theorem 3

In this section, we empirically verify Theorem 3 regarding two variables: the neural network's width, $W$, and the number of clean samples. The experiment is conducted with poisoned models trained over Trojan WM poisoned CIFAR-10 (poison rate: 20%, target label: 2).

| Model Name | # Parameters | W | Average Error Gap (%) |
|---|---|---|---|
| ResNet-18 | 11689512 | 512 | 0.35 |
| GoogLeNet | 6624904 | 832 | 0.27 |
| DenseNet-121 | 7978856 | 1024 | 0.24 |

Table 7: Empirical Error Gap with different widths ($W$) of the neuron networks. Each network is adopted and trained from scratch for 50 epochs and achieves the same level of ASR ($99.48 \pm 0.5\%$). We obtained the Error Gap using the original poisoning trigger (Trojan WM) after the model was defended by I-BAU and reported the results from an average of 5 runs.

Table 7 shows the empirical results of adopting different models with different widths. All the poisoned models are poisoned with Trojan WM attack using a poison rate of 20%. We sorted the models according to their maximum width ($W$) in Table 7. The Error Gap is obtained as the absolute value of the test error subtracted by the training error after conducting the defense. Based on the observation, the error gap is smaller as the model width grows, indicating better generalizability. Aligning with Theorem 3, the generalizability has a positive correlation with $W$.

| # Clean Samples | Average Error Gap (%) |
|---|---|
| 500 | 1.67 |
| 2500 | 0.71 |
| 5000 | 0.35 |

Table 8: Empirical Error Gap with different numbers of available clean samples. We adopted the poisoned ResNet-18 from Table 7 for this experiment. We obtained the Error Gaps using I-BAU defended models with different clean samples and reported the results from the average of 5 runs.

Table 8 shows the empirical results of adopting different numbers of clean samples during the I-BAU. As indicated from the results, a larger number of clean samples would lead to a smaller value of the Error Gap, which indicates a better generalization of unlearning effect from training to unseen data. Such results aligned with Theorem 3.

## A.3 Implementation Details and Complexity Analysis

I-BAU does not need to compute the second-order derivative directly. Instead, it is computed via implementing an approximation of the response Jacobian via an iterative solver (e.g., conjugated gradient algorithm (Rajeswaran et al., 2019) or fixed-point algorithm (Grazzi et al., 2020)) in limited rounds along with the reverse mode of automatic differentiation (Baur & Strassen, 1983; Griewank & Walther, 2008) by treating the problem as a linear system. Automatic differentiation in reverse mode is a widely used technique in modern deep learning packages such as Tensorflow and PyTorch (Baydin et al., 2018). This section gave the ablation study over the norm bound given in Algorithm 1 and the memory and time complexity analysis and comparisons.

### A.3.1 Ablation study on the $l_2$ norm bound

This section studies the impact of the preset $l_2$ bound's influence in the I-BAU unlearning scheme. We tested five different bounds to illustrate the effects, i.e., 0.5, 5, 10, 20, and Best Efforts, as shown in Figure 2. The settings of Best Efforts norm bound is that we do not include a norm constrained of the synthesized trigger as long as the trigger's value is within the image value range (from 0 to 1 in our case with float type images).

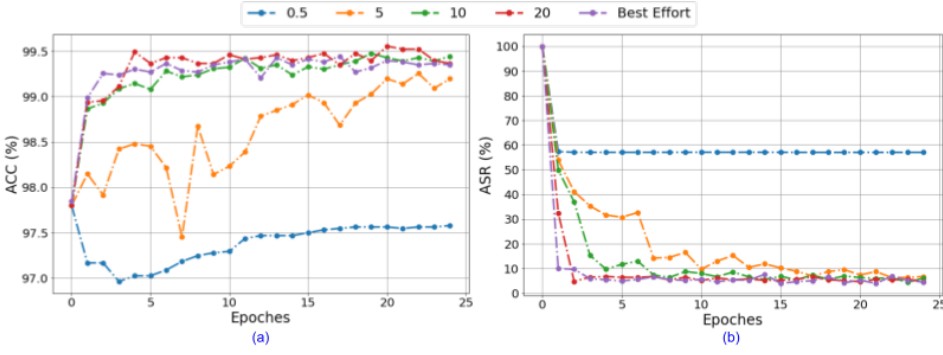

Figure 2: Evaluation of different $l_2$ norm bound would impact the ACC/ASR on mitigating Trojan WM attack on the GTSRB dataset. Each of the results listed is averaged from 5 independent runs using different random seeds.

The $l_2$ norm of launching the Trojan WM attack is 8.739 (measured by comparing with a zero matrix of the same size). As shown in Figure 2 a larger norm bound leads to a more robust and accurate synthesis of the potential trigger on the GTSRB. Especially the $l_2$ norm bounds that are greater than

the attack trigger's $l_2$ bound would lead to an effective defense in terms of low ASR and low impacts over the clean ACC. Based on Figure 2, we find that a large norm bound does not significantly impact over the clean ACC, namely the tread-off between the $l_2$ bound and the ACC drop is not substantial. In practice, when adopting I-BAU for backdoor defense, one is encouraged to adopt a large norm bound, thus encompassing more potential attacks.

### A.3.2 MEMORY COMPLEXITY ANALYSIS

Following Griewank (1993), we assume that the space complexity of computing $\nabla\delta(\theta) = -\left(\nabla_1^2 H(\delta(\theta), \theta)\right)^{-1} \nabla_{1,2}^2 H(\delta(\theta), \theta)$ via automatic differentiation is no more than twice the memory used when computing $\nabla H(\delta, \theta)$, which making our space complexity as $Mem(\nabla H(\delta, \theta))$. Recalling another popular class of methods to solve bilevel optimization—explicit gradient methods (Grazzi et al., 2020), whose memory complexity is $Mem(K \cdot T \cdot \nabla H(\delta, \theta))$ as they need to save the full computational graph during backpropagation, where $K$ is the number of rounds for adversarial unlearning, $T$ is the number of computations for the inner. In comparison, I-BAU is more memory efficient by adopting the implicit gradient via the iterative solver to approximate the computational graph without saving the whole graph.

| Input ($32 \times 32 \times 3$) |
|---|
| Conv2d $3 \times 3$ ($32 \times 32 \times 32$) |
| Conv2d $3 \times 3$ ($32 \times 32 \times 32$) |
| Max-Pooling $2 \times 2$ ($16 \times 16 \times 32$) |
| Dropout $(0.3)$ ($16 \times 16 \times 32$) |
| Conv2d $3 \times 3$ ($16 \times 16 \times 64$) |
| Conv2d $3 \times 3$ ($16 \times 16 \times 64$) |
| Max-Pooling $2 \times 2$ ($8 \times 8 \times 64$) |
| Dropout $(0.4)$ ($16 \times 16 \times 32$) |
| Conv2d $3 \times 3$ ($8 \times 8 \times 128$) |
| Conv2d $3 \times 3$ ($8 \times 8 \times 128$) |
| Max-Pooling $2 \times 2$ ($4 \times 4 \times 128$) |
| Dropout $(0.4)$ ($16 \times 16 \times 32$) |
| Flatten (2048) |
| Dense (C) |

Table 9: The target model details. The simplified VGG model contains three simplified VGG blocks, of which each contains two convolutional layers in each block. Here, we report the size of each layer.

### A.3.3 TIME COMPLEXITY ANALYSIS

Following our design of Algorithm 1 (total $K$ rounds), assuming using the fixed-point algorithm Grazzi et al. (2020) as the iterative solver with $\vartheta$ iterations for line 7, Algorithm 1, the time complexity would be $\tilde{O}(K \cdot \vartheta \cdot \tilde{O}(\theta))$, where $\tilde{O}(\theta)$ is the time complexity of training a neuronal network, $\theta$, via backpropagation for one epoch on the clean images used for unlearning (for most of the experiments, we used 5000 samples). In practice, we adopted $\vartheta = 5$, and for most of the one-target attack cases, $K = 1$ is enough to provide effective defenses (ASRs drop to random guessing rate). Below are some theoretical analyses and comparison of I-BAU with other state-of-art defenses listed in Table 6 regarding the time complexities:

- NC and TABOR require to go through all classes ($C = 10$ for the CIFAR-10 and $C = 43$ for the GTSRB), and each label requires a large number of steps ($K_1$ steps) of optimization to synthesize the trigger. Roughly their time complexity under the settings of limited iterations is $\tilde{O}(K_1 \cdot C \cdot \tilde{O}(\theta))$. In practice, $K_1 \cdot C$ is much larger than $K \cdot \vartheta$.
- DI incorporated an additional GAN to synthesis the trigger, assuming training and implementing the GAN is of the time complexity $\tilde{O}(\theta_{GAN})$, thus making the total time complexity roughly equals to $\tilde{O}(max(\tilde{O}(\theta_{GAN}), \tilde{O}(\theta)))$. In practice, the overhead of training a GAN trigger inspector is much expensive (estimated $300\times$ longer GPU time on the CIFAR-10) than training $\theta$.
- FP mitigates backdoor attacks via multi rounds ($K_2$ rounds) of pruning the network; in practice, FP requires more than 100 rounds of pruning (used half the number of samples for pruning, and the rest is for fine-tuning) to meet the stop requirements.
- NAD's time complexity is proportional to the number of epochs used to fine-tune the student and teacher models. As those two models share the same structure, we assume the time complexity of training them over the unlearning dataset is $\tilde{O}(\theta)$. Assuming teacher model training phase takes $K_3$ epochs, and tuning student model based on the teacher model takes $K_4$ epochs, then the total time complexity is $\tilde{O}((K_3 + 2 \times K_4)\tilde{O}(\theta))$. In practice, we adopted $K_3 = K_4 = 20$ according to the original work.

In conclusion, we find that theoretically, I-BAU is more efficient than other state-of-art defenses, and the theoretical results are aligned with the empirical observations over the average time taken effect over one-target attacks (see Table 6).

## A.4    EXPERIMENTAL DETAILED SETTINGS

The details of the simplified VGG model adopted in our paper are explained in Table 9. For each convolutional layer, we used batch normalization, and ELU is adopted as the activation function for each. We use Adam with a learning rate of 0.05 as the optimizer for poisoned models. The models are trained with 50 epochs over each poisoned dataset to converge and attain the results shown in the main text. Our experiment adopted ten NVIDIA TITAN Xp GPUs as the computing units with four servers equipped with AMD Ryzen Threadripper 1920X 12-Core Processors. Interestingly, the same experiments showed slower convergence (it takes more rounds to mitigate the backdoors) using GTX TITAN X and GTX 2080 TI. To reproduce the exact experimental results, we suggest considering adopting NVIDIA TITAN Xp GPUs for the experiments. PyTorch (Paszke et al., 2019) is adopted as the deep learning framework for implementations. For the settings of implementing the I-BAU, the inner and outer is conducted with iterative optimizers (SGD or Adam) with a learning rate of 0.1.

### A.4.1    ATTACKS DETAILS

We list the examples of the adopted backdoor attacks in this section. We incorporated eleven different backdoor attacks in this work. Figure 3 shows the examples from CIFAR-10 and the GTSRB before and after patched with different backdoor triggers. We adopted the same target label on the two datasets under one-trigger-one-target settings, as listed in Figure 3: *BadNets white square trigger targeting at label 8* (BadNets) (Gu et al., 2017), *Hello Kitty blending trigger targeting at label 1* (Blend) (Chen et al., 2017), $\ell_0$ *norm constraint invisible trigger targeting at 0* ($\ell_0$ inv) (Li et al., 2020a), $\ell_2$ *norm constraint invisible trigger targeting at 0* ($\ell_2$ inv) (Li et al., 2020a), *Smooth trigger (frequency invisble trigger) targeting at 6* (Smooth) (Zeng et al., 2021), *Trojan square targeting at 2* (Troj SQ) (Liu et al., 2018b), *Trojan watermark targeting at 2* (Troj WM) (Liu et al., 2018b). For both CIFAR-10 and the GTSRB poisoned models, we train the models with the entire training set (50000 samples for CIFAR-10, 39209 for the GTSRB) with the fixed 20% poison rate across all the experiments. For unlearning, the available clean data is sampled from both datasets' test set with a fixed size of 5000, where the remaining data (5000 for the CIAFR-10 and 7630 for the GTSRB) will be used to evaluate the unlearning efficacy (ACC and ASR).

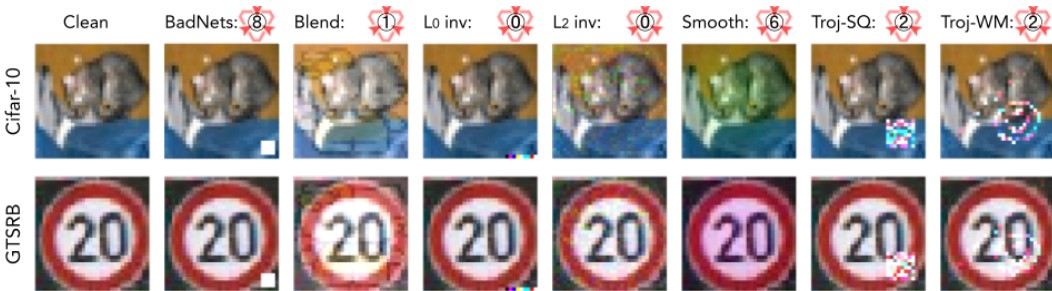

Figure 3: Datasets and examples of backdoor attacks that considered in the main text. We consider two different datasets in this work, namely, the CIFAR-10 dataset and the GTSRB dataset. Nine different backdoor attack triggers are included in the experimental part with one-trigger or multi-trigger attack patterns. Above, we show the target label used during the one-trigger attacks (e.g., badnets targeting at label 8) of each backdoor attack.

### A.4.2    BASELINE DEFENSES DETAILS

We compared I-BAU with six state-of-art backdoor unlearning defenses: *Neural Cleanse* (NC) (Wang et al., 2019), *Deepinspect* (DI) (Chen et al., 2019), TABOR (Guo et al., 2019), *Fine-pruning* (FP) (Liu et al., 2018a), *Neural Attention Distillation* (NAD) (Li et al., 2020b), and *Differential Privacy* training as a general robustness defense (DP) (Du et al., 2019). The detailed settings of comparison are provided as follows:

- **NC** is conducted following the same settings as the original work but only use the same 5000 samples as ours; for the outlier detection, we marked and unlearned all the detected triggers (Median Absolute Deviation (MAD) based on the generated trigger's norm, marked all the triggers whose mask MAD loss larger than 2 as detected).
- **DI** adopted model inversion technique for agnostic to the clean samples, yet made the defense's efficacy highly depend on the inversion technique. For a fair comparison, we feed the same 5000 samples, which are available to the other methods, to the GAN synthesizer in DI and obtains the final results; for the outlier detection, we marked and unlearned all the detected triggers (MAD based on the average loss for generating a trigger, marked all the triggers whose MAD loss larger than 2 as detected).
- **TABOR**'s settings follow the original work but with only 5000 clean samples being provided. (MAD based on the norm computation proposed in the original work to get rid of false alarms, marked all the triggers whose MAD loss larger than 2 as detected.)
- **FP** follows the suggestions of the original work, where we prune the network by supervising the ACC to drop to a certain percentage, i.e., 20%. This part's ACC is done by using 1000 samples from the 5000, and the rest 4000 clean samples are used to fine-tune the model to recover the ACC.
- **NAD**'s implementation follows the exact settings as the original work. The original work did not emphasize much over the preprocessing, and we used the same preprocessing following their open-sourced codes [1].
- **DP** follows the settings in the work that first mentioned use DP as a backdoor defense (Du et al., 2019), where we tuned the noise multiplier to attain universal effectiveness across all the considered attacks(50 for the CIFAR-10, 1.5 for the GTSRB).

## A.5   CASE STUDY ON NON-ADDITIVE BACKDOOR ATTACKS

As our fundamental formulation takes backdoor triggers as additive noise, in this specific section, we would like to evaluate the effectiveness of I-BAU towards non-additive backdoor attacks empirically. We selected four unique backdoor attacks/ settings to evaluate I-BAU towards non-additive attacks, which will be introduced as follows. 1) Semantical replacement (**SR**), which directly replaces the poison image with a piece of different semantical information. We designed this attack by directly changing all the poisoned images to an out-of-distributed 'Hallo Kitty' image with the poisoned label. SR should be considered as one of the worst-case attack scenario in practice, as its trigger is additional semantical information with a large norm bound; 2) **WaNet** (Nguyen & Tran, 2021) adopts a universal wrapping augmentation as the backdoor trigger. Under such a case, the backdoor trigger becomes a specific augmentation technique but not direct information insertion or addition. And WaNet has shown its ability to bypass some existing defense methods; 3) **IAB** attack (Nguyen & Tran, 2020) adopts autoencoder to learn and assign sample-specific noise to inputs to launch sample-specific backdoor attacks; 4) Hidden trigger (**HT**) attack (Saha et al., 2020) adopts a unique poisoning procedure using projected gradient descent to compute adversarial noise, which we consider as another example of a non-additive attack. We evaluate the effectiveness of I-BAU against the above four non-additive attacks on the CIFAR-10 dataset. We will illustrate their specific settings and results in the following parts of this section.

### A.5.1   TOWARDS MITIGATING SR

We first evaluate an extreme case where we replace the poisoned CIFAR-10 images with an out-of-domain 'Hallo Kitty' image. Such a procedure directly changes the semantic information of the poisoned data. We set the target label as '0'. Like the other evaluated attack, we replaced 20% of the non-target-class samples with the trigger kitty and set the label as '0'. As for launching the attack during test time, we evaluate the same 'Hallo Kitty' image exposed to the poisoned model and measure the attack success rate (in this case it becomes either 100% or 0%). The target model here adopted the small VGG16 introduced in our experiment. As for I-BAU, we adopted Adam optimizer and a learning rate of 0.1 and an unlimited $l2$ norm bound (instead, we crop the perturbation's value and restricted it to 0-1 as discussed in Appendix A.3.1). The results before and after are listed below in Table 10.

---

[1] https://github.com/bboylyg/NAD

| Clean ACC - Before | ASR - Before | Clean ACC - After | ASR - After |
|---|---|---|---|
| 83.82% | 100.00% | 82.12% | 0.00% |

Table 10: Empirical evaluation on I-BAU's effectiveness towards semantic replacement as backdoor attack.

As demonstrated in Table 10, within three rounds of I-BAU, we obtained a clean model with an acceptable ACC drop compared to the baseline. Interestingly, the extreme case directly inserts an additional semantical link between the poison kitty and the class '0', which can be interpreted as direct exposure of a specific training sample to the test set and interfere with the model generalizability (aka, overfit to a rare feature). Surprisingly, I-BAU can effectively mitigate such attack patterns. To our best knowledge, the above attack setting is never considered before. I-BAU also shows a potential path towards resolving model overfitting issues. We will leave such discussion to future work.

### A.5.2 TOWARDS MITIGATING WANET

WaNet is one of the famous invisible attack, instead of adopting an additive trigger, it adopts the same elastic transformation as the trigger of the attack. We directly downloaded the pre-trained poisoned PreActResNet18 on the CIFAR-10 from their work as the poison model to be adversarially unlearnt [2]. As for I-BAU, we adopted Adam optimizer and a learning rate of 0.0001 and an unlimited $l2$ norm bound. The results before and after are listed in Table 11, which indicates an effective defense with acceptable influence in the ACC.

| Clean ACC - Before | ASR - Before | Clean ACC - After | ASR - After |
|---|---|---|---|
| 94.36% | 99.62% | 92.86% | 10.80% |

Table 11: Empirical evaluation on I-BAU's effectiveness towards WaNet.

### A.5.3 TOWARDS MITIGATING THE IAB ATTACK

An emerging line of attack focuses on sample-specific attacks; here, we evaluate against IAB attack, which utilizes an autoencoder to learn and insert sample-specific triggers. We followed the same implementation as provided in the original work [3], with the following specific settings: dataset: CIFAR-10; target label:0; $\rho_b = \rho_c = 0.1$; model: PreActResNet18. One difference is that we loaded the CIFAR-10 dataset in a customized dataset format instead of the default (i.e., we used "torch.Tensor" loaded from "NumPy.array" with range $[0, 1]$, instead of "torch.Tensor" loaded from "PILimages" with range $[-1.99, 2.13]$). We did this to facilitate the implementation of I-BAU (which targeting at noises ranging from $[0, 1]$ for the current implementation). The results prior to and during the intervention are summarized in Table 12, indicating an effective defense with a low influence in the ACC.

| Clean ACC - Before | ASR - Before | Clean ACC - After | ASR - After |
|---|---|---|---|
| 87.28% | 99.20% | 86.46% | 9.58% |

Table 12: Empirical evaluation on I-BAU's effectiveness towards the IAB attack.

### A.5.4 TOWARDS MITIGATING HT

Finally, we evaluated the effectiveness of Hidden trigger (HT) backdoors (non-additive during poisoning), whose trigger inserting process is by directly resolving adversarial noise generated via projected gradient descent. We evaluated I-BAU with the CIFAR-10 random pairs attack settings from the original work (trigger_10, target: 8, source: 5, number of samples to generate PGD noise: 1500, number of poison in target class: 800, $\epsilon = 16$, optimization for generating poison: 0.01 with a decay rate of 0.95 every 2000 iterations). However, we found the original settings suffer from limitations in targeted ASR in our experiment, which is only "18.30%" (i.e., only drops the ACC after patching the trigger but have a relatively low chance leading to the target label). To enforce

---

[2] https://github.com/VinAIResearch/Warping-based_Backdoor_Attack-release
[3] https://github.com/VinAIResearch/input-aware-backdoor-attack-release

a successful targeted attack, we enlarged $\epsilon = 50$. During the fine-tuning process of [1], we only fine-tuned the clean model (ACC:"84.60") over the poison data, which resulted in a poisoned model with an ACC/ASR of "73.41/89.00". After adopting I-BAU for 20 rounds, the poisoned model's performance becomes "84.58/11.10", and with larger rounds (90 rounds), the performance can further be improved to "84.06/0.23", which indicates a robust and effective defense and an extra effect on recovering ACC.

### A.5.5 HIGHLIGHTS ON THE CASE STUDIES TOWARDS NON-ADDITIVE ATTACKS

The above results from the case study on using I-BAU to mitigate non-additive attacks highlight that although our formulation targets a universal pattern that most misled misclassifications in an additive way, I-BAU is empirically effective towards mitigating non-additive attacks. These emperical results have a great chance to lead to some exciting future works on theoretical analysis of the effectiveness of our proposed minimax formulation. We will open-source all incorporated attacks (at the moment, eleven attacks are incorporated, seven in the main text, and four non-additive case studies in the Appendix) and the pre-trained poisoned models. We will constantly check out the emerging attacks and look forward to seeing the first attack can evade our defense!

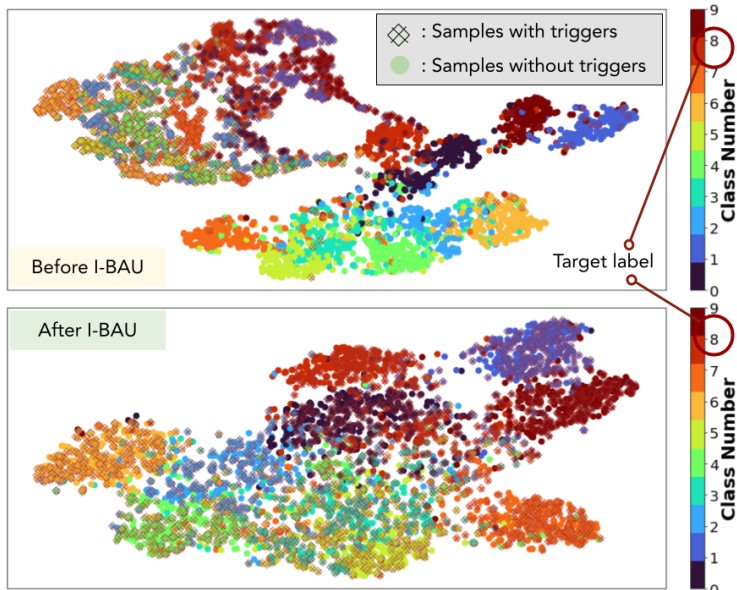

Figure 4: TSNE analysis of the output features (flatten layer output of the NN, with a size of $(N \times 2048)$) before and after backdoor unlearning using CIFAR-10 BadNets (targeting at 8) poisoned model. Each color in the color bar represents a different class from $(0, \ldots, 9)$. We mark the sample with triggers and without triggers as listed in the Figure.

### A.6 TSNE ANALYSIS ON UNLEARNING EFFECTS

The TSNE analysis of the feature extracted by the poisoned model and the unlearned model is shown in Figure 4. The model considered here is a BadNets poisoned model on the CIFAR-10 dataset (target label is 8), and the results listed in Figure 4 are before and after one round of I-BAU. Before the I-BAU, the poison model's extracted features for samples with/without triggers are disparate, even for samples originating from the same class. After the I-BAU, we can see that the unlearned model will map the samples patched with triggers back to their original classes (same color). Such results demonstrate the effectiveness of the backdoor unlearning from another perspective.

### A.7 ITERATIVE ILLUSTRATION ON MULTI-TARGET CASES

We show the iterative records of I-BAU mitigating more complicated attack cases, i.e., the all-to-all attacks and 7-trigger-7-target cases on the two evaluated datasets. We listed them here as they take

more rounds than one-trigger-one-target cases, which can usually be mitigated in a one-shot-kill manner.

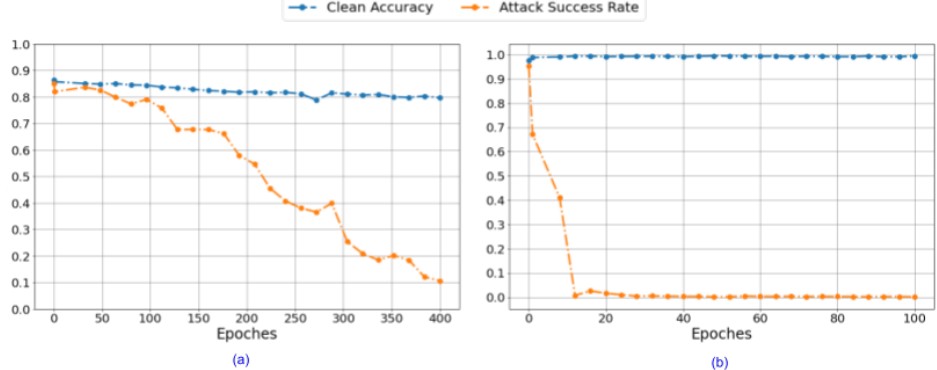

Figure 5: I-BAU iterative records under mitigating BadNets in all-to-all attack cases. (a). the results on the CIFAR-10 where took more rounds for the I-BAU to mitigate the attack successfully. (b). the results on the GTSRB manage to mitigate the triggers with fewer rounds and less impact over the ACC.

Figure 5 shows the results on countering the BadNets all-to-all attacks, where (a) is the results on the CIAFR-10 dataset, and (b) is the results on the GTSRB dataset. As shown in Figure 5 (a), although it takes more rounds than one-trigger-one-target cases to mitigate the attack, thanks to the accurate computing of the hyper gradient, the I-BAU did not impact much over the ACC. When the ASR drops below 10%, the unlearned model can still maintain an ACC above 80% (original ACC: 86.38). Meanwhile, in Figure 5 (b), we see that we can unlearn the BadNets trigger under all-to-all cases with even fewer rounds of I-BAU, and the unlearned model can maintain an ACC of around 99% during the entire unlearning procedure.

Figure 6 demonstrates the mitigation records of each iteration of I-BAU over the 7-trigger-7-target cases over the two evaluated datasets. Figure 6 (a) draws the details on the CIFAR-10 dataset, which took more than 200 rounds of I-BAU to mitigate all seven attacks. On the GTSRB, the mitigation of all seven triggers takes less time. And the model can maintain an ACC close to 99% during the entire unlearning procedure.

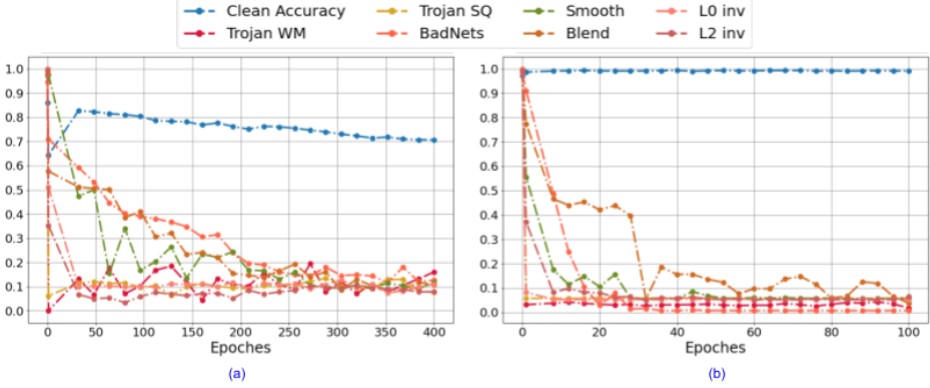

Figure 6: I-BAU iterative records under mitigating 7-trigger-7-target attack cases. (a). the results on the CIFAR-10 where took more rounds for the I-BAU to mitigate the attacks successfully. (b). the results on the GTSRB manage to mitigate the triggers with fewer rounds and less impact over the ACC.

Upon observation, there are triggers easier to be found by the I-BAU, e.g., Trojan WM and Trojan SQ, as they are optimized triggers, and thus easier to be found by the I-BAU. Such interesting observation might lead to new logic to consider while designing backdoor triggers (more optimized triggers might be easier to remove, as demonstrated).

