# OpenReview forum: "Adversarial Unlearning of Backdoors via Implicit Hypergradient"
_ICLR.cc/2022/Conference — ICLR 2022 Poster_

### Official Review · Reviewer_mJLa · 2021-11-02

**Correctness:** 3
**Technical Novelty And Significance:** 3
**Empirical Novelty And Significance:** 3
**Recommendation:** 6
**Confidence:** 3

**Main Review:**

1. In most cases, the proposed I-BAU is not the state-of-the-art method. For example, (I) in Table 1, I-BAU is not the best defense for 6 out of 8 Attacks, i.e. BadNets, l_0 inv, Smooth, Trojan SQ, Trojan WM, All to all; (II) in Table 3, I-BAU is not as good as NAD in terms of ASR; and (III) in Table 4, NC has lower ASR and higher/comparable ACC than the proposed I-BAU. Therefore, it is questionable if I-BAU is able to reduce the attack success rate to a very low level.

2. In page 6, only a simplified VGG model is used throughout the paper. So one question is whether the proposed method works well with different backbones, especially deeper networks.

3. In page 2, it is assumed that the norm of the backdoor pattern delta is bounded by C_delta. However, in real-world applications, the magnitude of this norm may be unknown, thus restricting the usage of the proposed method.

4. There is no empirical validation and justification of Theorem 1-3. For instance, in Theorem 3, the generalization bound can be measured by certain norms of weight matrices, the number of samples, Lipschitz constant, and so on. Hence, after a model is trained and fixed, it is possible to empirically justify if the theorems are true. It is also interesting to empirically investigate the bounds with regard to different weights (i.e. network architectures), number of samples, etc.


**Summary Of The Paper:**

The paper studies the backdoor attack problem and proposes a minimax formulation for the defense against adversaries. Theoretically, the paper analyzes the convergence bound and the generalization bound for the proposed method. Empirically, the paper compares efficacy, stability, sensitivity, and efficiency with several competitive methods.

**Summary Of The Review:**

One of the main issues is that the proposed method can hardly achieve the lowest ASR. Although I-BAU seems to be more stable, one may resort to the best defense for a corresponding attack if ASR has a high priority.

---

> ### Author Response · Authors · 2021-11-17
> **Response to Reviewer mJLa**
>
> We greatly value your feedback. Our responses are listed below.

---

> > ### Author Response · Authors · 2021-11-17
> > **Q4: [A practical way to select norm constraints]**
> >
> > **Q4: [A practical way to select norm constraints]** *(“$\ldots$ in
> > real-world applications, the magnitude of this norm may be unknown, thus
> > restricting the usage of the proposed method.”)*
> >
> > **A:** Note that there is a tradeoff between robustness and accuracy. A
> > larger norm tends to increase robustness but decreases accuracy. So, for
> > a practical way to select the norm constraint in the real world, we
> > suggest running a few ablation studies of the norm and picking the largest norm that meets the accuracy requirement.

---

> > > ### Comment · Reviewer_mJLa · 2021-11-29
> > > **Response**
> > >
> > > Thank you for your responses, which addressed my concerns. I will keep my positive score for this paper.

---

> > ### Author Response · Authors · 2021-11-17
> > **Q3: [Empirical validation of Theorem 1-3]**
> >
> > **Q3: [Empirical validation of Theorem 1-3] *(we have additional
> > results!)***
> >
> > **A:** (*We have added an empirical verification of Theorem 3 regarding the model width, $W$, and the number of clean samples, $n$. The results are affirmative, and we have included them in Appendix ``A.2.3``.*)
> >
> > | Model Name   | # Parameters |   $W$  | Average Error Gap |
> > |--------------|:------------:|:----:|:-----------------:|
> > | Resnet-18    |   11689512   |  512 |        0.35       |
> > | GoogLeNet    |    6624904   |  832 |        0.27       |
> > | DenseNet-121 |    7978856   | 1024 |        0.24       |
> >
> >
> > Regarding Theorem 3, we first added an evaluation setting considering
> > different models with different maximum widths, $W$ (see above). The
> > evaluation is done by using the one-trigger-one-target attack setting
> > from Section 6.1 with the Trojan WM trigger. For a fair comparison, all
> > the evaluated model is trained with their original structures with the
> > same optimizer and learning rate from scratch for 50 epochs; we observed
> > the same level of ASR across all the poison models ($99.48 \pm 0.5 \%$).
> > We evaluate the error rate gap using the Trojan WM after one round of
> > I-BAU. The error gap reported above is computed with the absolute value
> > of the test error minus the training error, and we report the average
> > results from 5 runs. Based on Theorem 3, a lower error rate gap
> > indicates a better generalization, aligning with the empirical results.
> >
> > | # Clean Samples | Average Error Gap |
> > |:---------------:|:-----------------:|
> > |       500       |        1.67       |
> > |       2500      |        0.71       |
> > |       5000      |        0.35       |
> >
> >
> > Then, we adopted ResNet-18 and evaluated how the number of clean
> > samples, $n$, would affect the generalizability. Following the settings
> > above, we report the average results from 5 runs. We find that the
> > larger the $n$ drops, the error rate Gap gets smaller, indicating
> > stronger generalizability. Which also aligned with the results and the
> > analyses of Theorem 3.
> >
> > The above results and analysis are also included in Appendix ``A.2.3``.

---

> > ### Author Response · Authors · 2021-11-17
> > **Q2: [Different backbones]**
> >
> > **Q2: [Different backbones] *(we have additional results!)***
> > *(“$\ldots$ whether the proposed method works well with different
> > backbones, especially deeper networks?”)*
> >
> > **A:** (*I-BAU works well with other backbones or larger models.*)
> >
> > We have added an evaluation regarding larger models with Trojan WM
> > poisoned CIFAR-10 targeting label 0 with a poison rate of 20% for
> > demonstration. The sizes of the parameters, baseline test ACCs trained
> > from scratch for 20 epochs, and the ACC/ASR after the I-BAU for one
> > round are listed in the table below.
> >
> > | Model Name   | # Parameters | Before |       | After |       |
> > |--------------|--------------|:------:|:-----:|:-----:|-------|
> > |              |              |   ACC  |  ASR  |  ACC  |  ASR  |
> > | Resnet-18    | 11219091     |  81.12 | 99.40 | 79.24 | 10.88 |
> > | DenseNet-161 | 28681000     |  83.28 | 99.86 | 80.56 | 10.28 |
> > | ResNext101   | 88791336     |  83.38 | 99.76 | 81.36 | 11.34 |
> >
> >
> > Noticing that the baseline ACCs are not as high as what we were expected
> > (by adopting those deeper models), for two reasons:
> >
> > -   The poison ratio is large, i.e., 20 percent of the data providing
> >     false gradients, thus impeding from a high ACC in the current
> >     setting (not the main factor).
> >
> > -   The models were trained from scratch for only 20 epochs, but we
> >     witnessed strong overfitting (main factor), we added drop-out layers
> >     (p=0.5) after each activation layer and tried our best to fine-tune
> >     them, but due to the time limit, the reported test ACCs are the best
> >     we can get.
> >
> > However, regarding the ASR, we can conclude such protection is working
> > well with larger and deeper models. We recorded that the I-BAU takes
> > longer than the simplified VGG to run (``15.2``s to ``19.8``s), but the defense
> > is still effective within one round. Noticing the other methods’
> > efficiencies are also dependent on the target model size; thus, the
> > other baseline’s (in Table 6) overhead is also proportional to the model
> > size, and our reported time will still be the lowest.

---

> > ### Author Response · Authors · 2021-11-17
> > **Q1: [Not always achieving the lowest ASR]**
> >
> > **Q1: [Not always achieving the lowest ASR]** *(“$\ldots$ it is
> > questionable if I-BAU is able to reduce the attack success rate to a
> > very low level.”, “$\ldots$ one may resort to the best defense for a
> > corresponding attack if ASR has a high priority.” )*
> >
> > **A:** Thank you for raising concerns about the ASR. Our ASR is close to
> > the random guess. We would like to clarify our experiment setting, which
> > can help understand why not attaining the lowest ASR (yet close to
> > random guess) is, in fact, not a concern. In our experiments, we patch
> > triggers to hold-out samples from all classes and use the patched
> > samples as our test set. We decided NOT to exclude the target class from
> > the test set for all the experimental settings. We did this to keep all
> > the settings the same and comparable, as in all-to-all and
> > multi-to-multi settings, we cannot remove the samples from the target
> > classes (e.g., in the all-to-all setting, all classes are target
> > classes, and removing samples from target classes would leave no clean samples).
> >
> > Following the above settings, if the ASR is close to random guessing
> > (10% for the CIFAR-10, 2.30% for the GTSRB), the defenses are considered
> > effective at a similar level (Page 7). For those comparison groups that
> > achieve far lower ASR than random guessing, e.g., Table 1, NAD’s
> > all-to-all results might be resulting from a high false-negative rate,
> > which also reflects in their lower ACC. Noting that the results on the ASR are not always “the lower, the better;” extremely low ASR can also
> > result from false positives as explained above. An ASR closer to the
> > random guess and meanwhile maintaining a high ACC is what should be
> > considered the better result. For example, NAD on mitigating the Trojan
> > WM attack on the CIFAR-10 achieved the lowest ASR, ``0.82``%. However, its
> > ACC is dropped to ``56.84``%. Meanwhile, our method achieved an ASR lower
> > than the random guess yet maintained the highest ACC (harmless to the
> > model usability).
> >
> > What else needs to be highlighted is that in practice, the triggers in
> > the training data are not known a priori. Hence, it is not possible to
> > preselect the best defense in accordance with the trigger. While some
> > baseline appears to achieve the lowest ASR for a certain trigger, the
> > fact that they are not robust against all triggers makes their practical
> > value questionable. On the other hand, the main advantage of our method
> > is the robustness to a range of diverse triggers with a slight
> > performance variation.
> >
> > Lastly, we want to clarify that due to the iterative nature of our
> > algorithm, we can always increase the number of iterations to reduce
> > ASR. For instance, as shown in the results from Figures 3 and 4, we can
> > always achieve an ASR that satisfies the user (as low as the user
> > desires) by considering the ASR-ACC trade-off. Hence, our approach is
> > still useful when the ASR is of top priority.

---

> ### Author Response · Authors · 2021-11-21
> **A friendly reminder of the rebuttal's conclusion**
>
> Respected reviewer mJLa,
>
> We'd like to express our gratitude once more for your constructive suggestions, which resulted in several interesting revision updates. *We've responded to each of your questions*. Hopefully, you'll find that they adequately address your concerns. Additionally, we'd like to know if you have any additional questions or require clarification before the rebuttal phase concludes. We would be delighted to address them in the rebuttal's revision.
>
> Best wishes,
>
> Authors of Paper2217

---

> ### Author Response · Authors · 2021-11-26
> **A friendly reminder that the discussion period's final stage is drawing to a close.**
>
> Respected Reviewer mJLa,
>
> We appreciate your positive feedback once again! We genuinely enjoyed your constructive suggestions, especially the empirical verification of the theoretical analysis, which helped make this work more polished. This letter is a friendly reminder that the discussion period's final stage is drawing to a close. If you happen to be down for any further discussion, this is the best time 😄.
>
> Regards,
>
> Authors of the Paper2217

---

### Official Review · Reviewer_u6du · 2021-11-02

**Correctness:** 3
**Technical Novelty And Significance:** 2
**Empirical Novelty And Significance:** 2
**Recommendation:** 6
**Confidence:** 3

**Main Review:**

Pros:
1. The idea of using implicit hypergradient in adversarial training is, to the best of my knowledge, novel and interesting.
2. The paper demonstrates that I-BAU is effective against multiple backdoor attacks in various settings (# of triggers and # of target labels)

Cons:
1. The paper does not cite a relevant work by J.Geiping et. al [1] which successfully adopts adversarial learning to defend against backdoor attacks. Also, I-BAU is not compared to the algorithm in [1].
2. The paper's clarity could be improved, in particular grammar. For example, there is a typo in the abstract (Implicit Bacdoor) and numerous typos in the main body (diffrencial prevacy, etc.). Also, some sentences are very confusing, for example in section 6 it is written "Can I-BAU effectively remove various backdoor triggers". I am confused with how the proposed approach should remove the backdoor triggers.


[1] What Doesn't Kill You Makes You Robust(er): Adversarial Training against Poisons and Backdoors

**Summary Of The Paper:**

The paper formulates adversarial training against backdoor poison attacks and proposes to use implicit hypergradient to solve the minimax problem instead of breaking it down into separate inner and outer optimization problems. The authors perform theoretical analysis of the algorithm and find convergence bound and generalization bound for the method. Also, the authors evaluate the proposed adversarial training routine, called I-BAU (Implicit Backdoor Adversarial Unlearning) in three attack settings:  (1) One-trigger-one-target attack, (2) One-trigger-all-to-all attack, (3) Multi-trigger-multi-target attack and compare with six defenses. For each attack setting, the authors test 7 different backdoor attacks. The results show that the efficiency of I-BAU is comparable to the best baseline for each of the attacks and is more time-efficient than other defenses. Finally, the authors show that I-BAU leads to a more stable training comparing to using universal adversarial perturbations in adversarial learning.

**Summary Of The Review:**

Although the empirical results for the method are appealing and the idea of using implicit hypergradient is interesting, the paper does not review previous work in adversarial training against backdoor attacks [1] which is extremely relevant and does not compare the proposed method with the existing algorithm. Also, the writing could be significantly improved. Because of these concerns, I vote for weak reject.

---

> ### Author Response · Authors · 2021-11-17
> **Response to Reviewer u6du**
>
> We value your feedback. Our responses are included below.
>
> **Q1:[Comparison with [1] What Doesn’t Kill You Makes You Robust(er)]** *(we have additional results!)(“$\ldots$ the paper does not review
> previous work [1] which is extremely relevant and does not compare the
> proposed method with the existing algorithm.”)*
>
> **A:** Thanks for bringing up this interesting reference [1]. We did not
> compare with this paper for the following reasons:
>
> 1.  Unlike what its title suggested, the paper [1] is, in fact, NOT
>     adversarial training (especially for backdoor) because the inner
>     optimization does not attempt to synthesize a trigger that optimizes
>     or approximately optimizes the inner loss. Instead, it patches
>     randomly sampled checkboard (Section C.3, Appendix, [1]).
>
> 2.  The paper is still at a preliminary stage [unpubilshed in any conoference]. We have reproduced the results [see L.1] but found it ineffective towards BadNets and Trojan WM attacks. So we decided to exclude it from comparison.
>
> [L.1]: [Implementation of
> [1]](https://drive.google.com/file/d/1zYpuUfTSIOAAt19bm5XPi-u2azqXkui-/view?usp=sharing)
>
>   ------------------ ------------------- ------- ---------------------------
> \
> **Q2: [Writings and confusions]** *(“$\ldots$ paper’s clarity could be
> improved $\ldots$”, “I am confused with how the proposed approach should
> remove the backdoor triggers.”)*
>
> **A:** (*We have improved the writing and better elaborated our delivery.*)
>
> We apologize for the typos, and we have proofread the paper again and
> improved the writing in the updated revision.
>
> For confusion on how we address the problem of removing backdoors, we
> provide an intuitive explanation for our approach here. Intuitively, our
> formulation encourages the model to produce stable predictions despite
> the fact that the input is patched with a trigger within a certain
> radius. Specifically, the inner optimization attempts to find the most
> powerful trigger that maximizes the prediction loss; then, the outer
> optimization attempts to impose the model to produce correct
> classification even when the most powerful trigger is patched onto the
> input. We hope this intuitive explanation can help clarify the mechanism
> of our backdoor removal approach.

---

> > ### Comment · Reviewer_u6du · 2021-11-26
> > **Response to Authors**
> >
> > Thank you for the clarifications.
> >
> > The response partially resolves my concerns, however to see that [1] is ineffective against BadNets and Trojan WM attacks, it would be great to see the results of these experiments. Could you please include some numbers on comparing [1]  against I-BAU on these attacks and implementation details?
> >
> > Thank you!

---

> > > ### Author Response · Authors · 2021-11-26
> > > **Settings and results for comparing [1] with I-BAU**
> > >
> > > We appreciate your response!
> > >
> > > ****
> > > **Settings:**
> > >
> > > For implementation details of [1], we followed the same settings for their defense against BadNets. Here is their original paragraph in the appendix, page 16, right bottom, [link](https://arxiv.org/pdf/2102.13624.pdf):
> > >
> > > >*'For backdoor triggers, we do not need to optimize and sample a random checkerboard pattern with a random rectangular shape within $l_0$ < 45 (overestimating the actual $l_0$ bound for a gray-box setting) as well as a random location. We sample such a patch for every class in the dataset and then apply them to randomly chosen pairs of classes, replicating the attack without knowing the targeted class.'*
> > >
> > > More specifically, for implementing their defense, regarding each different batch of input samples, we:
> > > 1. randomly split the batch into two equal-sized subsets, namely lower split and upper split according to [1];
> > > 2. We then randomly sample 10 checkerboards within $l_0$<45 at ten random locations for each class number in CIFAR-10;
> > > 3. We then randomly split the lower split into ten splits and poison each split with a randomly generated checkerboard and their specific target label;
> > > 4. Then we randomly split the upper split into ten splits and patch each splits with a specific random generated checkerboard, but without modifying the labels;
> > > 5. Finally, we concatenate the two splits and fine-tune the infected model over it.
> > >
> > > To make a fair comparison, we use the same settings from I-BAU on the number of clean samples, which is 5000. We deployed the same poisoned model we used in our paper, which are the simplified VGG models, with each being poisoned with a specific trigger using a poison rate of 20%. **Full details of the implementation can be found in the link from our last response, [L1]**.
> > > ****
> > > **Results:**
> > >
> > > Here are some numbers regarding the comparison of [1] with I-ABU; each reported number is averaged from 5 runs.
> > >
> > > | Attack Name 	|       	| Original 	|       	| [1] (one round) 	|       	| [1] (after 50 rounds) 	|       	| I-BAU (Ours) 	|
> > > |:-----------:	|:-----:	|:--------:	|:-----:	|:---------------:	|:-----:	|:---------------------:	|:-----:	|:-----:	|
> > > |             	|  ACC  	|    ASR   	|  ACC  	|       ASR       	|  ACC  	|          ASR          	|  ACC  	|  ASR  	|
> > > |   BadNets   	| 84.94 	|   98.28  	| 79.88 	|      90.28      	| 57.54 	|         90.76         	| 83.35 	| 12.30 	|
> > > |  Trojan WM  	| 84.92 	|   99.96  	| 79.96 	|      22.54      	|  46.5 	|         90.96         	| 83.58 	|  3.42 	|
> > >
> > > To conclude, regarding the two evaluated cases, [1]'s performance is erratic and ineffective, while harms a lot over ACC.
> > >
> > > On BadNets ($4 \times 4$ white block), [1] is ineffective. With more rounds of [1], the ASR still holds high, with the ACC already dropping to 57.54%. On defending trojan WM, at the very beginning (first round of [1]), it seems the defense starts to take effect, where the ASR drops to 22.54%. However, with more rounds of [1], the trigger's effect starts to recover, and the ASR will bounce back to 90.96%, with the ACC dropping below 50%. In comparison, I-BAU provides a more robust, effective, and performance-maintaining (low impact over ACC) defense.
> > >
> > > Such results also emphasize the **generalizability limitations** of [1] (*they only tested on BadNets checkerboard and Hidden trigger attacks, which uses random $4 \times 4$ blocks as triggers*). Based on the results, we would like to highlight further the importance of resolving the backdoor synthesis as an optimization problem (**random sampling checkerboards is a problematic and over-simplified heuristic**).
> > >
> > > ****
> > > Hopefully, our response can help resolve your concerns 😉.

---

> > > > ### Comment · Reviewer_u6du · 2021-11-29
> > > > **Response to Authors**
> > > >
> > > > Thank you for providing the numbers. My concerns were properly addressed, so I would like to increase my score to 6.

---

> > > ### Author Response · Authors · 2021-11-29
> > > **To Reviewer u6du**
> > >
> > > Respected Reviewer u6du,
> > >
> > > Today marks the conclusion of the discussion period. We're curious if our previous response addressed your concerns or if you require any more details 😉.
> > >
> > > Best wishes,
> > >
> > > Authors of Paper2217

---

> ### Author Response · Authors · 2021-11-21
> **A friendly reminder of the rebuttal's conclusion**
>
> Dear reviewer u6du,
>
> We'd like to express our gratitude once more for your recognition of the novelty of adopting implicit gradient in backdoor defense. *We've responded to each of your primary concerns*. Hopefully, you'll find that they address your concerns adequately. Additionally, we'd like to inquire whether you have any additional questions or require clarification prior to the conclusion of the rebuttal phase. We would be pleased to address them in the revision of the rebuttal.
>
> Best wishes,
>
> Authors of Paper2217

---

### Official Review · Reviewer_EmBg · 2021-11-02

**Correctness:** 3
**Technical Novelty And Significance:** 3
**Empirical Novelty And Significance:** 3
**Recommendation:** 6
**Confidence:** 3

**Main Review:**

**Strength**

(1) Overall, this paper is well-organized and easy to read. The proposed algorithm seems natural. Extensive experiments are conducted to illustrate the effectiveness and efficiency of the proposed algorithm. The comparison with other baselines also demonstrates superiority, which really convinces me.

(2) This research also includes some theoretical analysis, which is actually the missing part in current studies on backdoor attacks. This could be useful for future research.


**Weakness**

(1) Can the authors give a detailed description of how to implement I-BAU in practice and its time complexity? The reviewer is confused with the implementation and the efficiency, since the update of I-BAU in Eqn. (4) requires an inverse of a second-order derivative, which may not be supported directly by PyTorch/Tensorflow and may be extremely slow. However, the authors argue that the proposed I-BAU is more efficient than other baselines in Section 6.4.

(2) Is the sole difference between I-BAU and naive method (i.e., adversarial learning with universal perturbation) the implicit gradient in Eqn. (2)? Specifically, they both generate adversarial perturbation for each individual example within the sampled batch and add the perturbations together. Next, the naive method merely updates model parameters with the direct gradient, whereas I-BAU updates with the direct gradient plus an additional implicit gradient. Is the reviewer's understanding correct?

If so, can the authors explain why such an extra implicit gradient could stabilize the training and reduce ASR significantly in Section 6.2? This is not very intuitive.

Furthermore, the proposed I-BAU inherits the shortcoming of the universal perturbation (i.e., "the adversarial effect of individual perturbations may be canceled out by the addition operation" in the first paragraph of Section 4. Algorithm). As a result, part of the comments in the first paragraph of Section 4 might be unfair for the natural algorithm design.

(3) This paper assumes that the perturbation pattern is static and fixed for any examples upon being patched (see Minimux formulation for defense in Section 3). Unfortunately, some attacks are contrary to such an assumption. For example, the input-aware backdoor (IAB) attack [R.1] makes use of a dynamic trigger that varies across inputs. Can I-BAU still defend against IAB attack?

Furthermore, the formulation of Eqn. (1) actually requires an assumption, namely, the trigger pattern is static and fixed across inputs, in addition to the bounded norm constraint. As a result, the discussion advantages 2) in Section 3 appears to be incorrect.

[R.1] Tuan Anh Nguyen and Anh Tran. Input-Aware Dynamic Backdoor Attack. In NeurIPS, 2020.

(4) This paper only evaluates results with a simplified VGG, which only has low natural accuracy on CIFAR-10 (80%-86% in this paper, while the natural accuracy of a standard VGG usually achieves 92.64% [R.2] or 93.34% [R.3]), which brings difficult to determine whether the proposed method still works in larger models.
A large model may still be able to remember the backdoor behavior even after fine-tuning, since its large compacity. It is better to evaluate the performance across more architectures, such as ResNet-18 which has at least 93% of natural accuracy.

[R.2] https://github.com/kuangliu/pytorch-cifar

[R.3] https://github.com/bearpaw/pytorch-classification

(5) In practice, after downloading a model from the Internet, we can't tell whether it is backdoored or not. If the model is a benign one, would I-BAU hurt its performance?

(6) In adversarial training, the perturbation budget has a significant impact on robustness and natural accuracy. Since I-BAU is a variant from adversarial training, how does the ASR/ACC change with respect to varying norm constraints $C_{\delta}$?

(7) (Just a suggestion; no influence on my decision) When evaluating the ASR of models, it would be better to exclude the samples belonging to the target class. Because the perfectly robust model achieves 0% of ASR, it is much easier to compare ASR directly (i.e., the lower the ASR, the more robust the model). Otherwise, because the perfectly robust model still predicts the target class for poisoned samples whose ground truth is the target class,  it would have a non-zero ASR (e.g., 10% on CIFAR-10).

If the authors could solve some concerns mentioned above, the reviewer would reconsider the rating.

**Summary Of The Paper:**

This study investigates defense against backdoor attacks for models that have already been trained. It proposes, in particular, a min-max formulation for backdoor defense, in which the inner maximum seeks a powerful trigger that leads to a high loss, while the outer minimum seeks to suppress the "adversarial loss", so as to unlearn the injected backdoor behaviors. To solve the minimax, the authors also propose a method, Implicit Backdoor Adversarial Unlearning (I-BAU). In addition, the authors also provide theoretical analysis including the convergence bound and generalization bound. Extensive experiments demonstrate the effectiveness and efficiency of the proposed method.



**Summary Of The Review:**

Although the proposed method is natural and effective and the paper has many merits, there are some concerns including implementation details, some discussion in the paper, and lack of evaluation on more model architectures. The reviewer will recommend this paper if some concerns have been addressed.

---

> ### Author Response · Authors · 2021-11-17
> **Response to Reviewer EmBg**
>
> Your feedback is greatly appreciated. Listed below are our responses.

---

> > ### Author Response · Authors · 2021-11-17
> > **Q5: [Difference with the naive method] & Q6: [Clarification on including samples belonging to the target class]**
> >
> > **Q5: [Further clarification on the difference with the naive method]**
> > *(“Is the sole difference between I-BAU and naive method the implicit
> > gradient?”, “$\ldots$ the proposed I-BAU inherits the shortcoming of the
> > universal perturbation $\ldots$ ”)*
> >
> > **A:** (*We also adopt a global gradient for noise generation in I-BAU.*)
> >
> > Thanks for raising the confusion. There is another critical difference
> > we should have highlighted between the I-BAU and the naive design, which
> > lies in the strategy of solving inner optimization. Specifically, we did
> > NOT use universal perturbation to solve inner optimization; instead, we
> > computed one *global* gradient of inner loss function summed over a
> > batch of images and solved the inner iteratively with the *global*
> > gradient. By contrast, in universal perturbation, they computed the
> > gradient individually for each sample and then added the individual
> > gradients up as the perturbation. The individual gradients may be
> > canceled out by the addition operation. Our gradient, on the other hand,
> > proceeds to a direction that tends to be effective *globally* for all
> > images in the batch. It prevents the cancelation of the addition
> > operation of noises generated based on individual gradients. As a
> > result, more robust and consistent performance is achieved, as shown in
> > Figure 1.
> >
> >   ------------------ ------------------- ------- ---------------------------
> > \
> > **Q6: [Clarification on including samples belonging to the target class
> > during evaluation]** *(“$\ldots$ it would be better to exclude the
> > samples belonging to the target class.”)*
> >
> > **A:** Thanks for the suggestion. We used a test set containing samples
> > from the target class to keep all the comparison settings the same. In
> > particular, in all-to-all and multi-to-multi settings, we cannot remove
> > the samples from the target classes. For instance, in the all-to-all setting, all classes are target classes, and removing samples from
> > target classes would leave no clean samples.

---

> > > ### Comment · Reviewer_EmBg · 2021-11-19
> > > **Additional question**
> > >
> > > Thanks for the detailed feedback.
> > >
> > > The authors have addressed some of my concerns. The implementation and time complexity analysis seem to be convincing.  Now, I still feel confused about the gradient of inner maximization. The following two items should be equal in my opinion, while the authors claim they are different,
> > >
> > > (1) global gradient of inner loss function summed over a batch of images $\nabla \sum_{i\} L_i$
> > >
> > > (2) summation of per-sample gradients of the inner loss function $\sum_i \nabla L_i$
> > >
> > > Since  $\sum_i \nabla L_i = \nabla \sum_{i\} L_i$, I am afraid that the I-BAU also suffers from the cancelation of the addition operation.
> > >
> > > By the way, I want to clarify my suggestion in Q6: If a sample is attached by the trigger and its ground truth is the same as the target class of the trigger, it will be better to exclude it.

---

> > > > ### Author Response · Authors · 2021-11-19
> > > > **Further clarification on I-BAU with global gradient**
> > > >
> > > > Your prompt response and feedback are greatly appreciated.
> > > >
> > > >
> > > > ****
> > > > **Further clarification on I-BAU with global gradient:**
> > > >
> > > > We apologize for not being as clear as we intended. In fact, since (1) the individual gradients in universal perturbation are evaluated at different perturbations and (2) each individual gradient is clipped before being aggregated, the sum of individual gradients in universal perturbation is not equal to the global gradient computed by our algorithm.
> > > >
> > > > Specifically, assuming $P_{p, \xi}(\cdot)$ is the norm projection of the gradient regarding $p$-norm with a constraint $\xi$ (non-linear), and $\delta$ is the synthesized noise. Given an initialized noise, $\delta_0$, and a batch of data with size $N$:
> > > >
> > > > - **In universal perturbation**, the individual gradient associated with the first data is evaluated at the initial perturbation $\delta_0$:
> > > >
> > > >   $
> > > >   G_1 = \nabla_{\delta} L(x_1+\delta, \theta) |_{\delta=\delta_0}.
> > > >   $
> > > >
> > > >   The resulting perturbation is
> > > >
> > > >   $
> > > >   \delta_1 = P_{p, \xi}(\delta_0 + G_1).
> > > >   $
> > > >
> > > >   Note that the gradient for the second data point is computed at $\delta_1$ instead of $\delta_0$:
> > > >
> > > >   $
> > > >   G_2 = \nabla_{\delta} L(x_2+\delta, \theta) |_{\delta=\delta_1}.
> > > >   $
> > > >
> > > >   $
> > > >   \delta_2 = P_{p, \xi}(\delta_1 + G_2).
> > > >   $
> > > >
> > > >   $
> > > >   \vdots
> > > >   $
> > > >
> > > >   In universal perturbation, the perturbation of each data point is calculated sequentially; hence, the gradient for each data point in the batch is evaluated at different perturbations. The resulting perturbation is also highly dependent on the order of loaded data. Whatsmore, the norm ball projection is conducted $N-1$ times, resulting in the loss of large values regarding individual samples during the updates.
> > > > - **In I-BAU**:
> > > >
> > > >   $
> > > >   G = \sum_{i=1}^N \nabla_{\delta} L(x_i + \delta, \theta) |_{ \delta=\delta_0},
> > > >   $
> > > >
> > > >   where all the gradients are computed at $\delta=\delta_0$ and prevent the problem in the universal perturbation (performance fluctuation resulting from the dependence on the order of $x_i$). **Given the above derivation,  even without clipping, the sum of individual gradients is not equal to the global gradient as they are evaluated at different perturbations.** The clipping introduces additional nonlinearity:
> > > >
> > > >   $
> > > >   \delta = P_{p, \xi}(\delta_0 + G),
> > > >   $
> > > >
> > > >   where norm ball projection is only being adopted once, and large values of individual gradients would take effect and influence the generation of the synthesized trigger.
> > > >
> > > > ****
> > > > Regarding the suggestion in **Q6**, we would like to express our gratitude for the additional clarification. We acknowledge that doing so would facilitate comparisons between various defenses under the same settings. And such a setting will be included in the final version.
> > > > ****
> > > > In addition, since we have addressed all the major concerns mentioned in summary, we would like to kindly ask if you have any additional questions or require further clarifications. We would be more than happy to address them before the rebuttal ends.

---

> > > > > ### Comment · Reviewer_EmBg · 2021-11-20
> > > > > **Thank you for feedback**
> > > > >
> > > > > Thanks. It is much clearer now. I have no more questions, and I will reconsider my rating.

---

> > > > > > ### Author Response · Authors · 2021-11-26
> > > > > > **A friendly reminder that the discussion period's final stage is drawing to a close.**
> > > > > >
> > > > > > Respected Reviewer EmBg,
> > > > > >
> > > > > >
> > > > > > We appreciate your positive feedback once again! We genuinely enjoyed our discussion and your constructive suggestions regarding the time-complexity analysis, resulting in a more polished version of this work. This letter is a friendly reminder that the discussion period's final stage is drawing to a close. If you happen to reconsider a rating change or are down for a further discussion, this is the best time 😄.
> > > > > >
> > > > > >
> > > > > > Regards,
> > > > > >
> > > > > > Authors of the Paper2217

---

> > ### Author Response · Authors · 2021-11-17
> > **Q4: [More architechtures & sample-specific triggers]**
> >
> > **Q4: [More architectures & sample-specific triggers] *(we have additional results!)*** *(“$\ldots$ difficult to determine whether the
> > proposed method still works in larger models $\ldots$”, “$\ldots$ Can I-BAU still defend against IAB attack? ”)*
> >
> > **A:** (*I-BAU works on larger models, and surprisingly it also defeats the IAB attack.*)
> >
> > We first added an evaluation regarding larger models with Trojan WM
> > poisoned CIFAR-10 targeting label 0 with a poison rate of 20% for
> > demonstration. The sizes of the parameters, baseline test ACCs trained
> > from scratch for 20 epochs, and the ACC/ASR after the I-BAU for one
> > round are listed in the table below.
> >
> > | Model Name   | # Parameters | Before |       | After |       |
> > |--------------|--------------|:------:|:-----:|:-----:|-------|
> > |              |              |   ACC  |  ASR  |  ACC  |  ASR  |
> > | Resnet-18    | 11219091     |  81.12 | 99.40 | 79.24 | 10.88 |
> > | DenseNet-161 | 28681000     |  83.28 | 99.86 | 80.56 | 10.28 |
> > | ResNext101   | 88791336     |  83.38 | 99.76 | 81.36 | 11.34 |
> >
> >
> > Noticing that the baseline accuracies are not as high as what is being
> > reported from [R.2] and [R.3], they are worse than our simplified VGGs,
> > as two factors played roles in it:
> >
> > -   The poison ratio is 20%, i.e., 20 percent of the data providing
> >     false or wrong gradients, thus impeding from a high ACC in the
> >     current setting (not the main factor).
> >
> > -   The models were trained from scratch for only 20 epochs, but we
> >     witnessed strong overfitting (main factor), we added drop-out layers
> >     (p=0.5) after each activation layer and tried our best to fine-tune
> >     them, but due to the time limit, the reported test ACCs are the best
> >     we can get.
> >
> > However, regarding the ASR, we can conclude that such protection is also
> > effective toward larger and deeper models. We showed that the I-BAU
> > takes longer than the simplified VGG to run (``15.2``s to ``19.8``s), but the
> > defense is effective within one round. Noticing the other baseline’s (in
> > Table 6) overhead is also proportional to the model size, so our
> > reported time will still be the lowest.
> >
> > On sample-specific triggers, we tested the efficacy of I-BAU on
> > defending the IAB attack. We followed the same implementation as
> > provided in [R.1], with the following specific settings: *dataset:
> > CIFAR-10; target label:0; $\rho_b=\rho_c=0.1$; model: PreActResNet18*.
> > One difference is that we loaded the CIFAR-10 dataset in a customized
> > dataset format instead of the default (i.e., we used ``torch.Tensor`` loaded
> > from ``NumPy.array`` with range [0,1], instead of ``torch.Tensor`` loaded from
> > ``PILimages`` with range [-1.99, 2.13]). We did this to facilitate the
> > implementation of I-BAU (which targeting at noises ranging from [0,1]).
> > The baseline results after the IAB attack are as follows: ACC/ASR,
> > ``87.28/99.20``. After one round of the I-BAU, the ACC/ASR becomes
> > ``86.46/9.58``, which indicates a successful defense. Such results highlight
> > that although our formulation targeting at a universal pattern that most
> > mislead to misclassifications, indeed, it can also effectively mitigate
> > universal actions (say adopting the same GAN to mapping sample-specific
> > noises). These results would lead to some exciting future works on
> > theoretical analysis of the robustness of sample-specific backdoor
> > attacks.

---

> > ### Author Response · Authors · 2021-11-17
> > **Q3: [ASR/ACC and the norm constraints]**
> >
> > **Q3: [ASR/ACC and the norm constraints] *(we have additional
> > results!)*** *( “$\ldots$ would I-BAU hurt model’s performance?” ,
> > “$\ldots$ how does the ASR/ACC change with respect to varying norm
> > constraints?”)*
> >
> > **A:** (*Under desirable settings, our I-BAU does not harm the ACC (sometimes, it even increase the ACC). Larger norm results more accurate synthesis of the noise empowers better defense.*)
> >
> > The figure below is an additional experimental result to evaluate how
> > the change of norm constraint will impact the ACC/ASR using the GTSRB
> > dataset and a poisoned model poisoned with Trojan WM. We show the
> > results averaging from 5 runs of each experiment.
> >
> > [Figure Presented in Google Drive](https://drive.google.com/file/d/1HoE7s6jrFuIY9BEXanblS9GZSTOgkFbG/view?usp=sharing)
> >
> > As shown in the figure, a larger norm leads to a more robust and
> > accurate synthesis of the potential trigger on the GTSRB, thus:
> >
> > -   The unlearning is more effective, taking fewer rounds to fully
> >     remove the trigger;
> >
> > -   The synthesized trigger is more accurate, which impacts less on the
> >     clean ACC.
> >
> > In practice, the harm on ACC also depends on:
> >
> > -   How complicated the dataset is to learn (e.g., our method on the
> >     GTSRB can increase the ACC, yet on the CIFAR-10, the ACC would drop
> >     slightly).
> >
> > -   How strong is the trigger, i.e., a more potent trigger or a
> >     trigger’s original ASR closer to 100, is easier to be synthesized by
> >     the I-BAU, leading to less impact over the clean ACC.
> >
> > -   How many clean samples are available for the unlearning.
> >
> > In conclusion, we do not see much reduction of the model performance
> > (ACC) for I-BAU under desirable settings. As for more complicated tasks
> > (harder datasets, more stealthy triggers, less number of clean samples),
> > I-BAU is empirically shown to attain the best trade-off between model
> > performance (ACC) and defense efficacy (robustness and ASR) compared to
> > other state-of-art defenses (Table 1,2,3,4, and 5).

---

> > ### Author Response · Authors · 2021-11-17
> > **Q2: [Time complexity and comparisons]**
> >
> > **Q2: [Time complexity and comparisons]**
> >
> > **A:** (*We have added the time complexity and comparison in our paper.*)
> >
> > Following our design of Algorithm 1 ($K$ rounds), suppose we use the
> > fixed-point algorithm as the iterative solver with $\vartheta$
> > iterations for line 7, Algorithm 1, the time complexity of Algorithm 1
> > would be $\tilde{O}(K\cdot \vartheta \cdot \tilde{O}(\theta))$, where
> > $\tilde{O}(\theta)$ is the time complexity of training a neuronal
> > network via backpropagation for one epoch on the clean images used for
> > unlearning (for most of the experiments, we used 5000 samples). In
> > practice, we adopted $\vartheta=5$, and for most of the one-target
> > attack cases, $K=1$ is enough to provide effective defenses (ASRs drop
> > to random guessing rate). Compared to other methods shown in Table 6,
> > Section 6.4:
> >
> > 1.  NC and TABOR require to go through all classes ($C=10$ for the
> >     CIFAR-10 and $C=43$ for the GTSRB), and each label requires a large
> >     number of steps ($K_1$ steps) of optimization to synthesize the
> >     trigger. Roughly their time complexity under the settings of limited
> >     iterations is $\tilde{O}(K_1\cdot C \cdot \tilde{O}(\theta))$. In
> >     practice, $K_1\cdot C$ is much larger than $K\cdot \vartheta$.
> >
> > 2.  DI incorporated an additional GAN to synthesis the trigger, assuming
> >     training and implementing the GAN is of the time complexity
> >     $\tilde{O}(\theta_{GAN})$, thus making the total time complexity
> >     roughly equals to
> >     $\tilde{O}(max(\tilde{O}(\theta_{GAN}), \tilde{O}(\theta)))$. In
> >     practice, the overhead of training a GAN trigger inspector is much
> >     expensive (estimated $300 \times$ longer GPU time on the CIFAR-10)
> >     than training $\theta$.
> >
> > 3.  FP mitigates backdoor attacks via multi rounds ($K_2$ rounds) of
> >     pruning the network; in practice, FP requires more than 100 rounds
> >     of pruning (used half the number of samples for pruning, and the
> >     rest is for fine-tuning) to meet the stop requirements.
> >
> > 4.  NAD’s time complexity is proportional to the number of epochs used to fine-tune the student and teacher models. As those two models share the same structure, we assume the time complexity of training them over the unlearning dataset is $\tilde{O}(\theta)$. Assuming teacher model training phase takes $K_3$ epochs, and tuning student model based on the teacher model takes $K_4$ epochs, then the total time complexity is
> >     $\tilde{O}((K_3 + 2\times K_4)\tilde{O}(\theta))$. In practice, we
> >     adopted $K_3 = K_4 = 20$ according to the original work.
> >
> > Thus, I-BAU is considered more efficient, and the above analysis is
> > aligned with empirical results in Table 6. We have added the time
> > complexity analysis the Appendix ``A.3.2``.

---

> > ### Author Response · Authors · 2021-11-17
> > **Q1: [Implementaion might be difficult]**
> >
> > **Q1: [Implementaion might be difficult]** *(“$\ldots$ how to implement
> > I-BAU in practice $\ldots$”, “$\ldots$ may not be supported directly by
> > PyTorch/Tensorflow $\ldots$”)*
> >
> > **A:** (*We have open-sourced our implementations using PyTorch in the supplementary files, it's ready to run, and one can see our implementation taking effects within 10 seconds.*)
> >
> > In practice, with the approximated trigger $\delta_i$, it is easy to
> > implement an approximation of the response Jacobian via an iterative
> > solver (e.g., conjugated gradient algorithm [1] or fixed-point algorithm
> > [2]) in limited rounds along with the reverse mode of automatic
> > differentiation [3, 4].
> >
> > Automatic differentiation in reverse mode is a widely used technique in
> > modern deep learning packages such as Tensorflow and PyTorch [5]. We
> > will open-source our implementation of the I-BAU with the iterative
> > solver along with the reverse mode of automatic differentiation in
> > PyTorch (which can also be found in our supplementary files). We also
> > included some pre-trained poisoned toy models one can play with, the
> > code is ready to run, and an effective defense should be witnessed
> > within 10 seconds along with a GPU.
> >
> > What's more, further details on the implementation and time/ memory complexity analysis are presented in Section 4 and Appendix A.3.
> >
> > [1] Rajeswaran, Aravind, et al. “Meta-learning with implicit gradients.”
> > (2019).
> >
> > [2] Grazzi, Riccardo, et al. “On the iteration complexity of
> > hypergradient computation.” International Conference on Machine
> > Learning. PMLR, 2020.
> >
> > [3] Baur, Walter, and Volker Strassen. “The complexity of partial
> > derivatives.” Theoretical computer science 22.3 (1983): 317-330.
> >
> > [4] Griewank, Andreas, and Andrea Walther. Evaluating derivatives:
> > principles and techniques of algorithmic differentiation. Society for
> > Industrial and Applied Mathematics, 2008.
> >
> > [5] Baydin, Atilim Gunes, et al. “Automatic differentiation in machine
> > learning: a survey.” Journal of machine learning research 18 (2018).

---

> ### Comment · Reviewer_EmBg · 2021-11-30
> **Response to authors**
>
> Thank you for your feedback during the rebuttal. Since my concerns were almost addressed, I increased my score to 6 after reconsideration.

---

### Official Review · Reviewer_GDHf · 2021-11-03

**Correctness:** 4
**Technical Novelty And Significance:** 3
**Empirical Novelty And Significance:** 3
**Recommendation:** 6
**Confidence:** 4

**Main Review:**

1 . The min-max formulation of backdoor removal is quite intuitive, however, the perturbation constraint is a bit hard to quantify. The authors adopt a simple L2 norm constraint, which definitely makes it easier for computation, yet it also constrains the defense effect to only L2 norm bounded triggers. For example, I am not quite sure whether the proposed method can effectively handle the hidden trigger attack in [1]. Maybe extending the framework to also Linf norm would be beneficial.

[1] "Hidden trigger backdoor attacks." Proceedings of the AAAI Conference on Artificial Intelligence. Vol. 34. No. 07. 2020.

2 . The derivation to eq (4) is straightforward. The authors mentioned that “it can be implemented in a memory-efficient manner”, however, I didn’t find any description on how this memory-efficient manner was implemented? From eq (4), it seems that one still needs to compute the second-order derivatives of H, which can still take quite a large memory?

3 . The convergence results seem to directly follow (Grazzi et al., 2020) with similar assumptions and conclusions. I wonder what is the uniqueness of this result presented in the paper?

4.  The trigger removal experiments in Table 1 showed that the proposed algorithm is indeed more comprehensive, however, it does not achieve the best performances on many test cases, especially in terms of ASR.

5. In Table 5, where the authors showed the relationship between performances and the number of clean samples. While most baselines seem to achieve worse results when the number of clean samples decreases, yet the proposed method actually got slightly better ASR with a small portion of clean samples. This is actually quite counterintuitive. Will that be due to the random choice of clean samples? The authors might want to take a look into the details.

Typo: Above Table 5, “extrema” -> “extreme”


**Summary Of The Paper:**

In this paper, the authors proposed a minimax formulation for removing backdoors from a given poisoned model based on a small set of clean data. Unlike previous work, which breaks down the minimax into separate inner and outer problems, the proposed algorithm utilizes the implicit hypergradient to account for the interdependence between inner and outer optimization. The authors also theoretically analyzed its convergence and the generalizability of the robustness gained by solving minimax on clean data to unseen test data. Extensive experiments showed improved backdoor defense performances and less computation time on several backdoor attacks over various attack settings.

**Summary Of The Review:**

The paper is in general well written. However, there still exists some concerns towards the applicability of the proposed method as well as the uniqueness of the theoretical result. I think it is on the borderline. Depending on the authors' response, I may raise my score.

---

> ### Author Response · Authors · 2021-11-17
> **Response to Reviewer GDHf**
>
> We appreciate your comments. Our responses are included below.

---

> > ### Author Response · Authors · 2021-11-17
> > **Q3: [Uniqueness of theoretical results] and Q4: [Performance]**
> >
> > **Q3: [Uniqueness of theoretical results]** *(“$\ldots$ what is the
> > uniqueness of this result presented in the paper?”)*
> >
> > **A:** Thanks for raising the question on the convergence bound. We
> > uniquely formulate the backdoor defense problem as a bi-level optimization problem in order to leverage/adapt the convergence bounds
> > from (Grazzi et al., 2020) to the minimax settings.
> >
> > \
> > **Q4: [Performance]** *(“$\ldots$ does not achieve the best performances
> > on many test cases $\ldots$”, “$\ldots$ the proposed method actually got
> > slightly better ASR with a small portion of clean samples $\ldots$”)*
> >
> > **A:** Our ASR is close to random guessing. We would like to clarify
> > our experiment setting, which can help understand why not always
> > attaining the lowest ASR (yet close to random guess) is not a concern.
> > In our experiments, we patch triggers to hold-out samples from all
> > classes and use the patched samples as our test set. We decided NOT to
> > exclude the target class from the test set for all the experimental
> > settings. We did this to keep all the settings the same and comparable,
> > as in all-to-all and multi-to-multi settings, we cannot remove the
> > samples from the target classes (e.g., in the all-to-all setting, all
> > classes are target classes, and removing samples from target classes
> > would leave no clean samples). Following the above settings, if the ASR
> > is close to random guessing (``10%`` for the CIFAR-10, ``2.30%`` for the GTSRB),
> > the defenses are considered effective at a similar level (Page 7). For
> > those comparison groups that achieve far lower ASR than random guessing,
> > e.g., Table 1, NAD’s all-to-all results might be resulting from a high
> > false-negative rate, which also reflects in their lower ACC.
> >
> > Similarly, as the number of clean samples drops, our ASR’s drop is
> > explainable. It is not always the case that the lower ASR,
> > the better -  extremely low ASR can also result from false positives in
> > our settings, as explained above. Thus, the drop should not always be
> > interpreted as performance growth. One key factor resulting in the lower
> > ASRs in our methods is overfitting to the limited number of clean
> > samples, thus expecting a higher false negative, i.e., a lower ASR with
> > a lower ACC. As we explained above, an ASR close to the random guess
> > (``10%`` on CIFAR-10) but with a higher ACC might be better than an ASR
> > close to zero with a lower ACC. The reduction of clean samples affects
> > the ASR by 1) reducing the strength of synthesizing the trigger and 2)
> > increasing the level of overfitting. Baselines change in ASRs are for
> > the following reasons:
> >
> > 1.  NC, TABOR’s ASRs increase due to weaker synthesis.
> >
> > 2.  DI’s ASR drops due to strong overfitting.
> >
> > 3.  FP is not effective under the evaluated setting.
> >
> > 4.  NAD’s ASR is fluctuating due to overfitting.
> >
> > We have revised the paper to address your two questions.

---

> > > ### Comment · Reviewer_GDHf · 2021-11-21
> > > **Thank you for the response**
> > >
> > > I thank the authors for their detailed response. It has resolved my major concerns in the empirical performances. Although I think the theoretical part of this work is not that significant, I believe the overall paper quality is good, and therefore I would like to raise my score to 6.

---

> > > > ### Author Response · Authors · 2021-11-22
> > > > **Sincere thanks for the increase in rating & Highlighting other theoretical contributions**
> > > >
> > > > We want to express our gratitude once more for the remarks and the rating increase 😆
> > > >
> > > > The convergence bound of our backdoor removal algorithm is indeed largely built upon existing results (Grazzi et al., 2020). However, we would like to clarify that **the convergence bound is only part of our theoretical result**. In particular,
> > > >
> > > > - We also analyze the generalization of our formulation for both linear models and neural networks, which we believe are **critical to getting a fundamental understanding of the effectiveness of backdoor unlearning**.
> > > > - We analyze the Rademacher complexity for backdoor unlearning for both linear models and neural networks. To our best knowledge, **this is the first work theoretically exploring backdoor defense's generalizability**. The key insight of our proofs is to encapsulate the backdoor trigger as part of model parameter dimensions and calculate Rademacher complexity for the augmented hypothesis class. Although our proof for generalization bound is based on Rademacher complexity, this key insight can be potentially leveraged to prove generalization using other techniques, like the VC dimension.
> > > > - We also validate the generalization result's correctness and show that it aligns well with the experimental results (Section A.2.3, added during rebuttal).
> > > >
> > > > The theoretical examination of backdoor defense has long been a glaring omission from existing research. We present the **first generalization bound** for backdoor removal, and the results will aid in the development of more generalizable and effective backdoor defenses in the future.

---

> > ### Author Response · Authors · 2021-11-17
> > **Q2: [Why is I-BAU memory efficient?]**
> >
> > **Q2: [Why is I-BAU memory efficient?]** *(“$\ldots$ how this memory-efficient manner was implemented?”)*
> >
> > **A:** (*Further implementation details and memory complexity analysis are added.*)
> >
> > We apologize for the missing implementation details. I-BAU does not
> > compute the second-order derivative directly. Instead, it is computed
> > via implementing an approximation of the response Jacobian via an
> > iterative solver (e.g., conjugated gradient algorithm [1] or fixed-point
> > algorithm [2]) in limited rounds along with the reverse mode of
> > automatic differentiation [3, 4]. Automatic differentiation in reverse
> > mode is a widely used technique in modern deep learning packages such as
> > Tensorflow and PyTorch [5]. Following [6], we assume that the space
> > complexity of computing $\nabla \delta(\theta) = -
> > \left ( \nabla_{1}^2H(\delta(\theta),\theta)  \right )^{-1}
> > \nabla_{1,2}^{2}H(\delta(\theta),\theta)$ via automatic differentiation
> > is no more than twice the memory used when computing
> > $\nabla H(\delta,\theta)$, which makes our space complexity as
> > $Mem(\nabla H(\delta,\theta))$. Here, when we refer to
> > “memory-efficient,” we were comparing I-BAU with another popular class
> > of methods to solve bilevel optimization—explicit gradient methods [2],
> > whose memory complexity is
> > $Mem(K \cdot T \cdot \nabla H(\delta,\theta))$, where $K$ is the number
> > of rounds for adversarial unlearning, $T$ is the number of computations
> > for the inner. We have added the implementation details into Appendix
> > ``A.3``.
> >
> > [1] Rajeswaran, Aravind, et al. “Meta-learning with implicit gradients.”
> > (2019).
> >
> > [2] Grazzi, Riccardo, et al. “On the iteration complexity of
> > hypergradient computation.” International Conference on Machine
> > Learning. PMLR, 2020.
> >
> > [3] Baur, Walter, and Volker Strassen. “The complexity of partial
> > derivatives.” Theoretical computer science 22.3 (1983): 317-330.
> >
> > [4] Griewank, Andreas, and Andrea Walther. Evaluating derivatives:
> > principles and techniques of algorithmic differentiation. Society for
> > Industrial and Applied Mathematics, 2008.
> >
> > [5] Baydin, Atilim Gunes, et al. “Automatic differentiation in machine
> > learning: a survey.” Journal of machine learning research 18 (2018).
> >
> > [6] Griewank, Andreas. “Some bounds on the complexity of gradients,
> > Jacobians, and Hessians.” Complexity in numerical optimization. 1993.
> > 128-162.

---

> > ### Author Response · Authors · 2021-11-17
> > **Q1: [Applicability and norm bounds]**
> >
> > **Q1: [Applicability and norm bounds]** *(we have additional results!)*
> > *(“the perturbation constraint is a bit hard to quantify $\ldots$,”
> > “$\ldots$ whether the proposed method can effectively handle the hidden
> > trigger attack $\ldots$” )*
> >
> > **A:** (*Other norm-bounded attacks are already included. We did extra experiments regarding the Hidden Trigger backdoor, and we can successfully defeat them.*)
> >
> > Thanks for pointing out the concerns about the $l_2$-norm perturbation
> > constraint. In fact, we have already evaluated other norm-bounded
> > triggers beyond $l_2$, e.g., $l_0$ invisible ($l_0$ inv) triggers, blend
> > triggers, and trojan (Trojan WM and Trojan SQ) triggers (see Figure 2).
> > Though these triggers are specified under different norm bounds, all of
> > them can be bouned via a certain $l_2$-constraint. More precisely,
> > $l_2$-norm bound implies other norm bounds (see [L.1]). Thus, adopting
> > the $l_2$ norm as default in the formulation can still induce robustness
> > to other norm cases.
> >
> > Moreover, suppose it is known *a priori* that the attack model is
> > defined in terms of a non-$l_2$ norm. In that case, we can directly
> > extend the current minimax formulation by specifying the feasible set in
> > terms of the desired norm. Such extension is straightforward: it only
> > requires modifying the projected gradient descent for inner optimization
> > to a different projection region, which can be done by using existing
> > methods in optimization literature.
> >
> > For your concern about whether our method can effectively handle the
> > Hidden trigger backdoors in [1], we have performed extra experiments
> > during the rebuttal period, and the answer is affirmative. We evaluated
> > I-BAU with the CIFAR-10 random pairs attack settings from [1]
> > (trigger\_10, target: 8, source: 5, number of samples to generate PGD
> > noise: 1500, number of poison in target class: 800, $\epsilon = 16$,
> > optimization for generating poison: 0.01 with a decay rate of 0.95 every
> > 2000 iterations). However, we found the original settings suffer from
> > limitations in targeted ASR in our experiment, which is only ``18.30%``
> > (i.e., only drops the ACC after patching the trigger but have a
> > relatively low chance leading to the target label). To enforce a
> > successful targeted attack, we enlarged $\epsilon = 50$. During the
> > fine-tuning process of [1], we only fine-tuned the clean model (ACC:
> > ``84.60``) over the poison data, which resulted in a poisoned model with an
> > ACC/ASR of ``73.41/89.00``. After adopting I-BAU for 20 rounds, the poisoned
> > model’s performance becomes ``84.58/11.10``, and with larger rounds (90
> > rounds), the performance can further be improved to ``84.06/0.23``, which
> > indicates a robust and effective defense and an extra effect on
> > recovering ACC. One can find our implementations in [L.2].
> >
> > [L.1]: [Equivalence between different
> > norms](https://en.wikipedia.org/wiki/Norm_(mathematics))
> >
> > [L.2]: [On defeating Hidden Trigger Backdoors](https://drive.google.com/file/d/14AZRxq-NMtBWHA9Rr7toZ-FRfsqVNz-5/view?usp=sharing)

---

> ### Author Response · Authors · 2021-11-21
> **A friendly reminder of the rebuttal's conclusion**
>
> Respected reviewer GDHf,
>
> Once again, we'd like to express our appreciation for your insightful and thorough reviews. We have taken into account *each of your primary concerns*. Hopefully, you'll find that they adequately address your concerns. Additionally, we'd like to ask if you have any additional questions or require clarification before the rebuttal phase concludes. We would be delighted to address them in the rebuttal revision.
>
> Best wishes,
>
> Authors of Paper2217

---

### Author Response · Authors · 2021-11-18
**Rebuttal Summary**

We would like to express our heartfelt appreciation to all reviewers for their insightful comments and suggestions. We deeply appreciate that our work is recognized as well written (GDHf, EmBg), comprehensive (GDHf, mJLa), effective (EmBg, u6du), backed with theoretical analysis (EmBg), and novel (u6du). We have updated the following sections throughout the rebuttal, and we used the color "Orange" to indicate those changes.

------------------ ------------------- ------- ---------------------------

**Section 4**: Added details on implementation.\
**Algorithm 1**: We have specified the implicit gradient is approximated with an iterative solver.\
**Appendix A.2.3**: We have added an empirical verification and analysis of Theorem 3.\
**Appendix A.3**: We have added details about the implementation in PyTorch. Additional memory and time complexity analysis and comparison are also presented.\
**Overall**: Corrections have been made to the paper's grammar and typos.

------------------ ------------------- ------- ---------------------------

We have revised our paper to incorporate all of the additional experimental settings suggested in response to all of the insightful comments. Kindly notify us if anything remains unclear or if you have any further suggestions.

---

### Public Comment · ~Hossein_Souri1 · 2022-01-31
**Updating Related Work**

Thank you for your interesting paper. In our recently published paper [1], we have proposed a new clean label backdoor threat model that significantly improves the ASR. Please consider mentioning our work in your paper.

[1] [https://arxiv.org/abs/2106.08970](https://arxiv.org/abs/2106.08970)

---

### Public Comment · ~Yi_Zeng3 · 2022-02-06
**Post Rebuttal (Final Version) Summary**

We want to express our gratitude to all reviewers and PCs for their priceless suggestions and comments! Based on the remarks of the PCs, we have updated the following sections in the final version in response to the discussion over norm-bound settings and 'non-additive attacks.'

------------------ ------------------- ------- ---------------------------

**Appendix A.3.1**: We have added an ablation study discussing the norm bound in Algorithm 1. We empirically observed that a large norm bound would not affect the ACC much, but a small bound might limit the effects towards large-normed triggers. We have included our suggestion that when adopting I-BAU, one should set the norm bound as large as possible (can be as large as the value range of the image pixels). \
**Appendix A.5**: We have added four more experiments considering four different non-additive attacks. It turns out that I-BAU is of excellent generalizability and empirically effective towards those non-additive attacks. We even included an extreme case where we directly changed all the poisoned images to a non-related 'Hallo Kitty' image (directly semantical information replacement) with the poison label.
------------------ ------------------- ------- ---------------------------

Overall, we would like to highlight that in the future, further theoretical analysis of the certified effectiveness of our proposed formulation would be an exciting direction. As it turns out, empirically speaking, I-BAU is not only effective towards additive triggers but also towards novel lines of backdoors that adopt autoencoders or augmentations to achieve sample-specific attacks (non-additive and non-universal-pattern).

We welcome all kinds of further discussions. Kindly notify us if anything remains unclear or if you have any suggestions 😊

---

> ### Public Comment · ~Xitong_Gao1 · 2022-09-07
> **Question related to the proposed bilevel optimization**
>
> Thanks for your interesting publication, I have a question regarding the bilevel optimization problem:
> $$
> \theta^\star = \arg\min_{\theta} H(\delta^\star(\theta), \theta),
> \text{where~}
> \delta^\star(\theta) = \max_{\|\delta\| \leq C_\delta} H(\delta, \theta)
> $$
> Since the inner objective is bounded and maximized, the optimality sits on the boundary of $\|\delta\| \leq C_\delta$, and $\nabla_x H(x, \theta)$ thus may not be equal to $0$ at its optimality $x = \delta^\star$. This violates the first order stationarity condition assumed by the implicit function theorem for (3) to work. I wonder how this is addressed in your algorithm?
>
> Thanks again and I appreciate your response in advance.
>
> Kind regards,

---

### Decision · Program_Chairs · 2022-01-20

**Decision:**

Accept (Poster)

**Comment:**

This paper investigates defense against backdoor attacks for models that have already been trained. It proposes, in particular, a min-max formulation for backdoor defense, in which the inner maximum seeks a powerful trigger that leads to a high loss, while the outer minimum seeks to suppress the "adversarial loss", so as to unlearn the injected backdoor behaviors. To solve the minimax, the authors also propose a method, Implicit Backdoor Adversarial Unlearning (I-BAU). In addition, the authors also provide theoretical analysis including the convergence bound and generalization bound. Extensive experiments demonstrate the effectiveness and efficiency of the proposed method.

The proposed method is interesting and the implementation is nice. Overall, there is a fundamental flaw in the formulation: if the trigger is not additive (where there are many such examples of poisoning attacks that are not additive) this approach should fail completely. Not having experiments that discuss such triggers that are not additive is a significant flaw in the presentation of the paper. Another flaw is that the trigger is assumed to be small norm. Unlike adversarial examples attacks (at test time), there is no reason for backdoor triggers to be of small norm. Given that the defense critically relies on these two flawed assumptions, and the extent of how the proposed algorithm is sensitive to these assumptions are not properly addressed in the experiments, this paper is on the border line.